# Ensemble-based snow depth data assimilation for a multi-layer snow scheme over the European Arctic

Åsmund Bakketun[1,2], Jostein Blyverket[1], and Malte Müller[1,2]

[1]Development Centre for Weather Forecasting, Norwegian Meteorological Institute, Oslo, Norway
[2]Department of Geosciences, University of Oslo, Oslo, Norway

**Correspondence:** Åsmund Bakketun (asmundb@met.no)

**Abstract.** Reliable estimates of Earth system conditions are important for weather forecasting, hydrological modelling and their downstream applications. Both real-time prediction systems and historical reanalyses use a combination of observations and physical laws embedded in numerical models to generate gapless and accurate estimates of weather, climate and hydrological conditions. Data assimilation systems merge information from model estimates and observations in an objective way, accounting for their respective uncertainties. In this work we present a regional reanalysis system, focusing on the land surface component. The system uses a multi-layer snow model together with the ensemble-based Local Ensemble Transform Kalman Filter (LETKF) data assimilation scheme. The system is run for a 4 year period over the European Arctic, assimilating in situ snow depth observations. Evaluation of the new snow depth analysis showed reduced errors compared to existing products and positive impact of the data assimilation over the domain. Furthermore, a significant difference in total accumulated snow water was seen over the domain, implying a potential impact on downstream hydrological applications. The ensemble correlations between the total snow depth and the multivariate control vector indicated that the ensemble was able to represent snow compaction processes. The LETKF is thus able to account for processes which are often neglected in snow depth data assimilation. The system presented in this study allows for future extensions, including other types of observations and analyses beyond snow variables.

## 1 Introduction

Accurate weather and climate estimates are crucial for a wide range of applications, including advancing our knowledge of the Earth system. In mountain regions and high latitudes, seasonal snow cover significantly influences land-atmosphere interactions. This is due to its high albedo, which reflects more sunlight compared to snow-free ground and its unique thermal properties (Gong et al., 2004). Moreover, realistic initial conditions of surface and snow variables play a key role in improving atmospheric predictions (de Rosnay et al., 2014). Seasonal snow cover also impacts local infrastructure and stores substantial amounts of water, which are released during spring melt (Viviroli et al., 2007; Sturm et al., 2017; Croce et al., 2018). Accurate estimates of snow cover and mass are also vital for managing hydropower stations and issuing timely flood warnings (Casson et al., 2018; Li et al., 2019; Magnusson et al., 2020).

Reanalysis products offer consistent time series of meteorological conditions on both global (Hersbach et al., 2020) and regional scales (Schyberg H. et al., 2021). By integrating vast amounts of observational data into numerical model simulations, they generate estimates of numerous parameters. Among these, global atmospheric reanalyses like ERA5 (Hersbach et al., 2020) and MERRA-2 Gelaro et al. (2017) have become foundational tools in Earth system research.

The land surface component of numerical weather prediction models is often simplified compared to state of the art models used in other communities (Fisher and Koven, 2020). Land surface models can be run stand-alone using atmospheric input from a forecast or reanalysis product. This allows for efficient testing of new configurations and production of datasets tailored for specific needs (Arduini et al., 2019; Zsoter et al., 2022). Several reanalyses feature a stand-alone "spin off" prioritizing improved representation of land surface processes. One such example is ERA5-Land, derived from ERA5. It incorporates a more advanced surface model and has a higher spatial resolution compared to the ERA5 dataset (Muñoz-Sabater et al., 2021). Despite the improved representation of the land surface, several challenges remain to be solved. Cao et al. (2020) found warm bias in ERA5-Land soil temperature in high latitudes affecting permafrost estimation, potentially due to snow density errors. Clelland et al. (2024) found no advantage of ERA5-Land relative to ERA5 over Siberia for a range of climate variables. Kouki et al. (2023) used satellite retrievals of snow variables to demonstrate improved spatial snow cover estimates in ERA5-land, but overestimated snow water equivalent particularly in high elevation areas compared to ERA5.

Regional reanalyses add value to the global reanalyses by providing higher resolution products over limited geographical areas (Køltzow et al., 2022). Notable examples include the Copernicus European and Arctic regional reanalysis products CERRA (Ridal et al., 2024) and CARRA (Schyberg H. et al., 2021), among others. They use the ERA5 global product as lateral boundary conditions and optimize observation usage and parametrization over Europe and the European Arctic region. Nevertheless, these products are based on simple land surface schemes which are not able to describe important processes. Monteiro et al. (2024) configured a multi-layer land surface model and improved the representation of seasonal snow in a numerical weather prediction system similar to the one used in the production of CARRA. While a reported evaluation of snow properties is missing for the CARRA dataset, it is likely to share similar deficiencies as the single layer snow scheme in the last mentioned study.

An essential component of reanalysis systems is data assimilation. Data assimilation is the method of combining model predictions and observations objectively based on their respective uncertainties. For snow analysis, observations often include in situ networks of snow depth measurements, satellite observations of reflectance and microwave radiance (De Lannoy et al., 2012; de Rosnay et al., 2014; Charrois et al., 2016; Micheletty et al., 2022; Gichamo and Draper, 2022). Furthermore, land surface temperature products could potentially be useful of improving snow estimates, as seen in synthetic experiments (Alonso-González et al., 2023). Sentinel-1 backscatter data have also shown promising results for both snow depth estimates and downstream river discharge (Brangers et al., 2024). Data assimilation schemes vary from direct insertion, which disregards the respective uncertainties (Hedrick et al., 2018), optimal interpolation techniques using static prescribed uncertainties (Brasnett, 1999) and ensemble-based schemes where the uncertainties are deducted from the ensemble covariances. Variants of the Ensemble Kalman Filter (EnKF) (Evensen, 2003; Tippett et al., 2003) and the particle filter (Magnusson et al., 2017; Cluzet et al., 2021; Alonso-González et al., 2023) are popular in snow data assimilation. While the EnKF have an underlying

assumption of normally distributed errors, the particle filter is relaxed on this constraint. However, the EnKF has shown to be robust also when the Gaussian assumption is not met (Katzfuss et al., 2016). With the EnKF, the analysis ensemble is a linear combination of the first guess ensemble. It is thus important that the linear assumption holds for the ensemble, particularly in multivariate analysis, otherwise it could lead to inconsistent or unrealistic states. A hypothetical example could be a two member ensemble where one member has old snow with high density and the other has fresh snow with low density. The ensemble-mean (example case of linear combination) would not represent any of the two conditions, nor a realistic snow pack for that time. Since the particle filter resamples the most likely states and does not apply linear combinations, it is not subject to this issue. However, it is more vulnerable to ensemble collapse. For the above example, the analysis could result in two copies of the same member. In general, the particle filter requires a larger number of members, but could represent diverging trajectories in contrast to the standard EnKFs.

In this work, we implement a regional land reanalysis system using recent developments of a multi-layer snow model together with an ensemble-based data assimilation scheme. Through evaluation over northern Scandinavia, we assess whether a more advanced system can enhance estimates of snow conditions compared to those using simplified schemes. Understanding the differences between simplified and more advanced schemes is valuable for the development of future reanalysis and weather prediction systems.

The configuration of the system and reference datasets are described in Sect. 2. Results and evaluation are presented in Sect. 3 and discussed in Sect. 4. Conclusion is given in Sect. 5.

## 2 Methods and data

In the following section, a brief description of the reference system (CARRA) is given, including details about its snow data assimilation scheme (2.1). Subsequently, the section presents the land surface model (Sect. 2.2), data assimilation method (Sect. 2.3), the ensemble generation and conditioning methods (Sect. 2.4-Sect. 2.5), a description of the observation and validation data (Sect. 2.6) and finally the experimental setup in Sect. 2.7.

### 2.1 Copernicus Arctic Regional Reanalysis (CARRA)

The CARRA dataset is produced by the convection-permitting numerical weather prediction model system HARMONIE-AROME (HIRLAM–ALADIN Research on Mesoscale Operational NWP in Euromed–Application of Research to Operations at MEsoscale) (Bengtsson et al., 2017). The dataset covers the area shown in Fig. 2, with a grid point spacing of 2.5 km. As we focus on the snow component of the reanalysis, we guide the reader to the full system documentation (Schyberg H. et al., 2021) for details concerning the remaining components.

The CARRA system uses the externalized surface (SURFEX) (Masson et al., 2013) as lower boundary in the model integration. The soil and snow is modelled by the force restore version of Interactions between Soil, Biosphere and Atmosphere (ISBA) (Noilhan and Mahfouf, 1996; Calvet et al., 1998; Decharme et al., 2011), where two soil layers represent rapid (hourly) and slow (daily) variations in temperature to provide fluxes back to the atmospheric component Noilhan and Mahfouf (1996).

The snow is represented by a single layer scheme (Douville et al., 1995). This snow model represents the snowpack with snow water equivalent, snow density and albedo as prognostic variables.

In the CARRA system, in situ snow depth measurements and a binary satellite snow cover product are assimilated to update snow water equivalent of the model. The in situ snow depth observations are converted to snow water equivalent using monthly climatological snow densities from a lookup table. The satellite snow extent data is converted to snow water equivalent pseudo observations following a set of rules. If the satellite reports "no snow", the pseudo observation yields $0 \ \mathrm{kg \ m^{-2}}$; if the satellite reports "snow", the pseudo observation is set equal to the model background value in the observation point. However, if the model exceeds $25 \ \mathrm{kg \ m^{-2}}$, "snow" observations are discarded, and if the model exceeds $100 \ \mathrm{kg \ m^{-2}}$, "no snow" observations are discarded. Furthermore, the snow analysis is not allowed to add snow on snow free grid cells if the surface temperature is above the melting-point of water. Such rules can be considered to represent some confidence in the model state as the direct insertion method does not account for that. The pseudo observation snow water equivalent value is then treated as an in situ observation. The satellite product and assimilation method are reported in Homleid and Killi (2014). The in situ measurements (including pseudo observations from satellite) are interpolated horizontally (from point measurements to a 2D field) based on distance dependent correlation functions with the optimal interpolation (OI) scheme (Brasnett, 1999). While the spatial correlation length is equal for in situ and pseudo observations from satellite, they are weighted differently relative to the background field with observation error ratios $\sigma_o = 1.0$ and $\sigma_o = 1.6$ respectively. The snow water equivalent is finally inserted directly into the model replacing the background field, snow density and snow albedo are not modified in the analysis but kept constant.

## 2.2 Land surface model

Production of the regional reanalysis presented in this study consists of two main components. A land surface model to cycle the model snow states between analysis times, and the assimilation scheme that ingests the observations into the model state variables. To model the time evolution of the snow state, we use the ISBA model (Noilhan and Mahfouf, 1996; Calvet et al., 1998; Decharme et al., 2011) within the SURFEX framework (Masson et al., 2013). The ISBA model is set up using two patches (low and high vegetation), the multi-layer diffusion soil scheme (Decharme et al., 2011), explicit snow scheme (Decharme et al., 2016) and the explicit canopy option (Boone et al., 2017; Napoly et al., 2017) for the high vegetation patch.

We have also adopted some of the configurations in Monteiro et al. (2024). In their study, they found that heat flux from the uppermost soil layer to the lowermost snow layer was too large in the case of low snow fractions leading to too rapid snow melt. They proposed a workaround by reducing the heat flux in these situations which we included in our system. We used soil clay, sand and organic carbon dataset from soilgrids (Hengl et al., 2017).

Using two patches means that within a grid cell, two independent mass and energy budgets are computed, adapted to the vegetation type. The resulting soil and snow states thus represent different sub grid conditions.

We use the ISBA explicit snow model with the default 12 layers, adding information about the vertical structure of the snow compared to the single layer model. The prognostic variables are snow water equivalent, snow density, snow heat content, albedo and snow age.

The land surface model is driven by atmospheric variables including precipitation (snow and rain), radiation (direct short-wave, diffuse shortwave and longwave), air temperature and humidity, surface pressure and wind. In this work, the forcing data is obtained from CARRA (Schyberg H. et al., 2021). The CARRA dataset has proven to perform better compared to the ERA5 product for both temperature and wind particularly in areas with complex topography (Køltzow et al., 2022; Box et al., 2023). The forcing data are interpolated spatially to the model grid using bilinear interpolation and SURFEX internally interpolates

the hourly values to the 10 minute model time step using linear interpolation.

## 2.3  Data assimilation

Data assimilation aims to bring the model state closer to the true unknown state based on observations. The corrected state, called the analysis ($\mathbf{x}^a$), can be expressed as the original state, often referred to as the background state ($\mathbf{x}^b$), and a correction term,

$$\mathbf{x}^a = \mathbf{x}^b + \delta\mathbf{x}. \tag{1}$$

Here the correction ($\delta\mathbf{x}$), referred to as the increment, is typically a function of the innovation vector ($\mathbf{y}^o - \mathbf{y}^b$) where $\mathbf{y}^o$ represents the observations, and $\mathbf{y}^b = h(\mathbf{x}^b)$ is the model state in observation space, with $h$ being a nonlinear observation operator.

    According to Helmert et al. (2018), the most commonly used method for snow data assimilation in numerical weather

prediction systems is the optimal interpolation (OI) method (e.g., Brasnett, 1999; Liston and Hiemstra, 2008; de Rosnay et al., 2014; Li et al., 2022). OI uses predefined structure functions to correlate observations with grid point values based on horizontal and vertical distances, often modelled with Gaussian curves (Gaspari and Cohn, 1999; Brasnett, 1999). When snow depth measurements are used to correct snow water equivalent and snow density, the solution is not unique. It is common practice to assume that the snow density is constant and only update the snow water equivalent (see references above). With more

advanced snow models, like ISBA explicit snow, the state vector becomes significantly larger compared to single layer models. With a 12 layer configuration, the full control vector consists of 49 prognostic variables at every grid point patch (snow water equivalent, snow density, snow age and snow heat, each times 12 layers plus snow albedo). Distributing the analysis increments between all the different control variables can thus be challenging. Brangers et al. (2024) chose to distribute the snow depth increments proportional to the background layer thicknesses, thus assuming the profile of the snow is unchanged. However, so

called flow-dependent data assimilation schemes can objectively optimize the analysis state by taking the current conditions (errors of the day) into account and thus narrow down the possible solution space and potentially updating the layers that actually are most likely to explain the observed condition. The EnKF (Evensen, 2003) uses an ensemble of model realizations to represent the background error covariance matrix and it is widely used in snow data assimilation (Slater and Clark, 2006; Durand and Margulis, 2007; Lannoy et al., 2010; Kumar et al., 2017; Brangers et al., 2024). In the Kalman Filter, the increment

is written

$$\delta\mathbf{x} = \mathbf{K}(\mathbf{y}^o - \mathbf{y}^b) \tag{2}$$

where $\mathbf{K}$ is called the Kalman gain matrix, and is defined as

$$\mathbf{K} = \mathbf{P}\mathbf{H}^T(\mathbf{H}\mathbf{P}\mathbf{H}^T + \mathbf{R})^{-1} \tag{3}$$

where $\mathbf{P}$ is the background error covariance matrix, $\mathbf{H}$ the linearized observation operator and $\mathbf{R}$ is the observation error covariance matrix. Furthermore, Lannoy et al. (2010) show that the EnKF gain matrix is written

$$\mathbf{K} = \text{cov}[\mathbf{x}, h(\mathbf{x})][\text{cov}[h(\mathbf{x}), h(\mathbf{x})] + \mathbf{R}]^{-1}. \tag{4}$$

Here the error covariance matrices are computed implicitly through the ensemble covariances.

In geophysical applications, the state vector has high dimensionality. For example, the CARRA domain has $1000 \times 800$ grid points which for a model with 12 vertical levels and 5 physical variables results in a control vector of size $\sim 5 \times 10^7$. Large ensembles can thus be computationally expensive. Consequently, small ensembles have to be used and sampling noise is inevitable. The sampling noise could lead to spurious correlations between variables (Leutbecher, 2019). Essentially, the spurious correlations result in increments based on unrealistic relationships between the observation and the control vector. This behaviour is avoided by limiting the impact of observations outside their representative area, referred to as localization. Localization can be applied by tapering the background covariance matrix, which means that the covariances between distant grid points are forced to zero. Another technique is the so called local analysis (Sakov and Bertino, 2011), where the data assimilation is split into smaller regions and only nearby observations are used.

Hunt et al. (2007) proposed a localized version of the EnKF, solving the filter equations individually at each grid point. Compared to the example above, the control vector is now only $12 \times 5 = 60$ elements. At each point, relevant observations (typically within a radius) are selected and used in the assimilation and inflation of observation errors as function of distance is often used to smoothly reduce observation impact over increasing distances. Due to the manageable use of computational resources and the flexibility regarding observation types, we have chosen their implementation of the local ensemble transform Kalman filter (LETKF). The LETKF has shown promising results in both atmospheric and surface data assimilation systems (Shin et al., 2016; Gastaldo et al., 2018; Seo et al., 2021; Nerger, 2022; Lee et al., 2024). The LETKF update equations for a grid point are:

$$\mathbf{x}^a = \bar{\mathbf{x}}^b + \mathbf{X}^b \mathbf{w}^a \tag{5}$$

$$\mathbf{w}^a = \mathbf{W}^a + \bar{\mathbf{w}}^a \tag{6}$$

$$\bar{\mathbf{w}}^a = \tilde{\mathbf{P}}^a (\mathbf{Y}^b)^T \mathbf{R}^{-1} (\mathbf{y}^o - \bar{\mathbf{y}}^b) \tag{7}$$

$$\mathbf{W}^a = [(k-1)\tilde{\mathbf{P}}^a]^{1/2} \tag{8}$$

$$\tilde{\mathbf{P}}^a = [(k-1)\beta^{-1}\mathbf{I} + (\alpha \circ (\mathbf{Y}^b)^T \mathbf{R}^{-1})\mathbf{Y}^b]^{-1} \tag{9}$$

where $\mathbf{x}$ represents the ensemble control vector, $\mathbf{X} = \mathbf{x} - \bar{\mathbf{x}}$ the ensemble anomalies (or perturbations), bar represents ensemble-mean, $a$ and $b$ indicate analysis and background respectively, $\mathbf{w}$ is the transformation weights between the background and the analysis, $\mathbf{Y}^b$ represent the ensemble perturbations of the observation equivalent. $\tilde{\mathbf{P}}$ and $\mathbf{R}$ are the error covariance matrices.

$\beta$ controls the inflation of the background error covariance matrix. $\alpha$ is a location dependent weight vector used to inflate $\mathbf{R}$ using element wise multiplication. Each element corresponds to an observation in $\mathbf{y}$. In line with structure functions used in the CARRA OI scheme, we use a Gaussian curve as function of distance for $\alpha$,

$$\alpha = \exp\left(-2(\frac{d_h}{r_h})^2 - 2(\frac{d_v}{r_v})^2\right), \tag{10}$$

where $d_h$ and $d_v$ are horizontal and vertical distances, respectively, between observations and the grid point, $r_h$ and $r_v$ are prescribed impact distances. Note that $\mathbf{y}$ holds the observations selected for the grid point. Despite that the assimilation is independent in each grid point, the same observations could be used in several close grid points depending on the specified impact distances and localization function for $\mathbf{R}$. Close grid points using the same observations thus obtain the same weights ($\mathbf{w}$) only modified by the localization weights $\alpha$. The resulting increments are connected to neighbouring points through the spatial structure of the background ensemble perturbations. This is illustrated in Fig. 1 f) where the observation is marked by a star and the correlations (indicated by colour shading) are proportional to the increments for the surrounding area. For spatially homogeneous ensemble perturbations, the spatial structure of the increment would be Gaussian centred on the station.

Diefenbach et al. (2023) shows that the analysis equation for the ensemble-mean $\bar{\mathbf{x}}^a$ can be written as

$$\bar{\mathbf{x}}^a = \bar{\mathbf{x}}^b + (k-1)^{-1}\mathbf{X}^a\mathbf{Y}^{aT}\mathbf{R}^{-1}(\mathbf{y}^o - \bar{\mathbf{y}}^b). \tag{11}$$

This is the common form of the Kalman Filter where $(k-1)^{-1}\mathbf{X}^a\mathbf{Y}^a\mathbf{R}^{-1}$ corresponds to the gain matrix $\mathbf{K}$ mapping the innovations $(\mathbf{y}^o - \bar{\mathbf{y}}^b)$ in observation space to increments $(\mathbf{x}^a - \mathbf{x}^b)$ in model space. Investigating the elements of $\mathbf{K}$ is important for understanding the filter performance. We recognize that $\mathbf{X}^a\mathbf{Y}^{aT}$ is proportional to the ensemble correlation between the model space state and the observation space state. The ensemble correlation between a model variable ($i$) and an observed quantity ($j$) is given by

$$r_{ij} = \frac{\mathbf{X}^i\mathbf{Y}^{jT}}{\sqrt{\mathbf{X}^i\mathbf{X}^{iT} + \mathbf{Y}^j\mathbf{Y}^{jT}}}. \tag{12}$$

In situ observations of snow depth are usually placed in flat open areas away from high vegetation. Since the surface model used in this study explicitly represents low and high vegetation one could choose to assume that the observation was more representative for the patch with low vegetation. However, there are some factors that complicates this approach. First, the low vegetation patch is not defined for all grid points. Second, not all observations are placed sufficiently away from high vegetation to only represent that patch. We consider this to be out of scope for this work. To compute the model state in observation space, we use the following observation operator:

$$y^b = h(\mathbf{x}^b) = \sum_{i}^{2}\sum_{j}^{12}\gamma_i\frac{w_{i,j}^b}{\rho_{i,j}^b} \tag{13}$$

where $w$ and $\rho$ represent the snow water equivalent and snow density at the closest model grid point respectively and $\gamma_i$ is the patch (open land or forest) fraction. The index $i$ represents the model patch and $j$ the snow model level.

## 2.4 Ensemble perturbations

For ensemble based data assimilation methods, the ensemble perturbations $\mathbf{X}$ and $\mathbf{Y}$ are used to represent the uncertainty of the model estimate. They also describe the relationships between locations and between variables which map innovations to increments as shown above. It thus require care when generating the ensemble to ensure realistic relationship between the variables. The uncertainty can be sampled by perturbing forcing data, model state variables, model parameters or a combination of these (Slater and Clark, 2006; Lannoy et al., 2010; Liu et al., 2011; Kumar et al., 2014, 2017; Blyverket et al., 2019; Draper, 2021; Seo et al., 2021; Brangers et al., 2024). The benefit of only perturbing the forcing data is that the resulting model states are consistent with the model physics and the relationship between variables is as realistic as the model. However, processes that are unresolved by the model, for example redistribution by wind in our model, will not be represented by the ensemble by only perturbing the forcing data. Since we in this work introduce a multivariate control vector using the univariate observation vector, we only perturb the forcing data (rain, snow, longwave radiation, shortwave radiation, air temperature) to reduce the chance of introducing sampling noise. We follow the above studies and use cross-correlation between forcing variables to ensure realistic forcing conditions, temporal auto-correlation of the noise and spatially correlated noise (Reichle and Koster, 2003; Durand and Margulis, 2007; Lannoy et al., 2010). The details about the perturbation parameters are found in Table 1. The cross-correlations shown in Table 1 are computed from one year of forcing data. Spatial and temporal correlation lengths are provided in Table 4.

**Table 1.** Perturbation Methods and Parameters for Forcing Variables

| Variable | Perturbation Method | Standard deviation $\sigma$ | Rain | LW | SW | Snow | Ta |
|----------|---------------------|------------------------------|-------|-------|-------|-------|--------|
| Rain | Multiplicative | 0.5 (fractional) | 1.000 | -0.066 | 0.177 | 0.091 | 0.123 |
| LW | Additive | 30 W/m$^2$ | -0.066 | 1.000 | -0.246 | -0.061 | 0.156 |
| SW | Multiplicative | 0.3 (fractional) | 0.177 | -0.246 | 1.000 | 0.078 | 0.493 |
| Snow | Multiplicative | 0.5 (fractional)) | 0.091 | -0.061 | 0.078 | 1.000 | -0.128 |
| Ta | Additive | 0.5 °C | 0.123 | 0.156 | 0.493 | -0.128 | 1.000 |

Precipitation forcing is usually perturbed by multiplying the field with noise from a log-normal distribution to avoid positive precipitation bias in the case of no precipitation (see references above). This method is not able to represent the possibility of precipitation if the original dataset has zero precipitation. Maggioni et al. (2012) used an advanced error model for satellite precipitation products to approach this issue and saw good results for soil moisture fields. However, it requires calibration with satellite and a radar reference which is not generally available. In this work we introduce a novel approach to improve the spatial representation of precipitation uncertainty. By assuming that precipitation fields in the forcing data have an error in horizontal placement, we can introduce uncertainty in areas surrounding such fields. We refer to the method as remapping and is based on advection using a random vector field. The approach builds on ideas from Le Coz et al. (2019), where precipitation fields are remapped to better fit observations. An example of the remapping method is shown in Fig. 1. In this case we show a precipitation field from the CARRA dataset (Fig. 1a) and several realizations after remapping b-d). The ensemble correlations

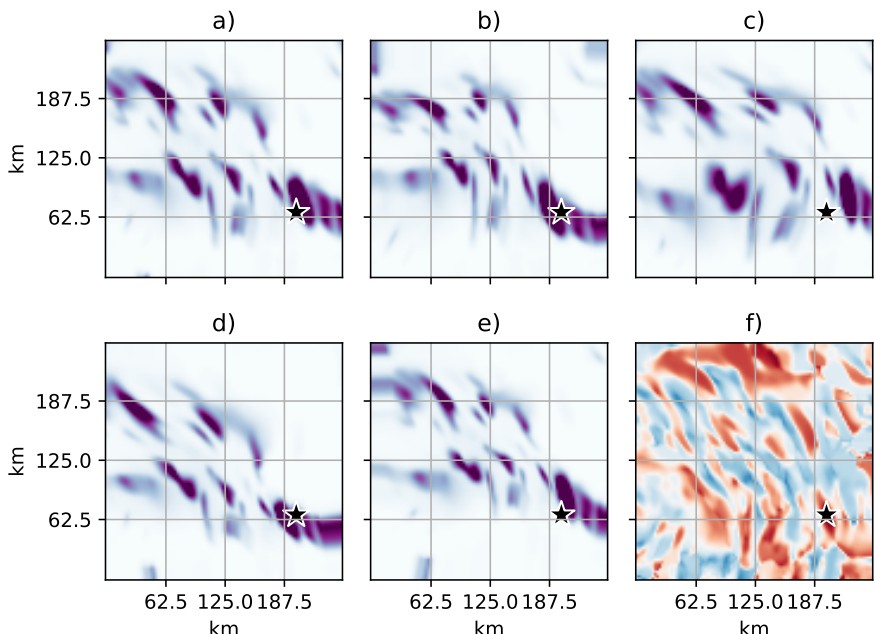

**Figure 1.** Illustration of remapping. a) shows original field (rain fall), b-e) shows the ensemble members after remapping. f) shows ensemble correlations between the point indicated by a star and the respective surrounding grid points.

shown in Fig. 1f) shows how an observation at the location indicated with a star can influence the surrounding area based on the current conditions. A detailed formulation of the method is found in Appendix A. We evaluated the method by comparing occurrences of snowfall at observation locations. For snowfall greater than zero the unperturbed snowfall dataset had a hit rate of 83 %, while the ensemble had 91 %. For snowfall greater than 5 $\mathrm{cm}$ (snow depth) over 24 hours the deterministic hit rate was 21 % while the ensemble hit rate was 56 %.

## 2.5  Conditioning of ensemble states

In the SURFEX model, snow water equivalent is defined in all model points. In grid points with no snow, the snow water equivalent is zero and the other prognostic variables (density, age, albedo and heat) are undefined. In order to include these grid points in the analysis and potentially add snow to snow free members, undefined values need to be initialized. While some predefined values can be used, e.g. corresponding to freshly fallen snow, such simplifications can quickly introduce nonlinear relationships between ensemble members. For example, during melting season, one member becomes snow free while the other members still have old and very dense snow. A different approach is to use a sample from the members containing snow to initialize zero snow members. Even though this method might not produce a representative spread for the snow, it avoids the filter to produce unrealistic analyses. In this work we initialized zero snow members with the ensemble-mean of members

**Table 2.** Valid ranges for snowpack quantities

| | |
|---|---|
| snow water equivalent $w$ | $0 \leq w$ |
| snow density $\rho$ | $50 \leq \rho \leq 917.3$ |
| snow temperature (diagnostic) $T$ | $200 \leq T$ |
| snow heat $h$ | $h < 0$ |

**Table 3.** Observation sets with corresponding acronyms, total number of stations and total number of observations during the experiment period

| Set | Assimilated in | Acronym | num. stations | num. observations |
|---|---|---|---|---|
| A | CARRA-Land-Pv1 | OBS-ONLY-Pv1 | 178 | 122590 |
| B | CARRA | OBS-ONLY-CARRA | 111 | 108652 |
| C | CARRA and CARRA-Land-Pv1 | OBS-BOTH | 144 | 124416 |

with snow. We use the ensemble-mean because it will have a neutral impact on the analysis ensemble spread. If all members are snow free, the filter is unable to add snow.

After the data assimilation is performed, the analysed state is the best estimate given that the assumptions of the filter hold.
However, it might not be physically consistent, nor within the model bounds. The result can thus cause instabilities in the preceding model integration For example, negative snow water equivalent can in theory occur in the analysed state. In the data assimilation system, we use the ensemble-mean approach as mentioned above for all snow related variables given that a member violates any of the conditions in Table 2. This method prioritizes stability of the model integration over optimal performance of the filter.

## 2.6 Validation and observation data

In situ snow depth observations used in the data assimilation are obtained from the Meteorological Archival and Retrieval System (MARS) of the European Centre for Medium-Range Weather Forecasts (ECMWF). The observation dataset is not identical to the one used in CARRA, as CARRA uses additional local observations and a different quality control. Thus, we can bin observations into three sets. Let $\hat{A}$ be the observations used in CARRA-Land-Pv1 and $\hat{B}$ the observations used in CARRA. Further let $C = \hat{A} \cap \hat{B}$ be the observations used in both systems, $A = \hat{A} - \hat{B}$ the stations only used in CARRA-Land-Pv1 and $B = \hat{B} - \hat{A}$ the stations only used in CARRA. In the following we will denote dataset $A$ as "OBS-ONLY-Pv1", $B$ as "OBS-ONLY-CARRA" and $C$ as "OBS-BOTH". In addition, snow water equivalent from 6 snow pillow stations (Tollan, 1970; Egli et al., 2009) are used to evaluate the unobserved (by the assimilation system) snow water equivalent state. In CARRA, satellite derived snow cover maps are converted to in situ pseudo observations and assimilated in a similar way as real in situ snow depth. We considered to assimilate satellite snow cover in this study, but chose to focus on point observations. We suggest that satellite snow cover products are included in a follow up study.

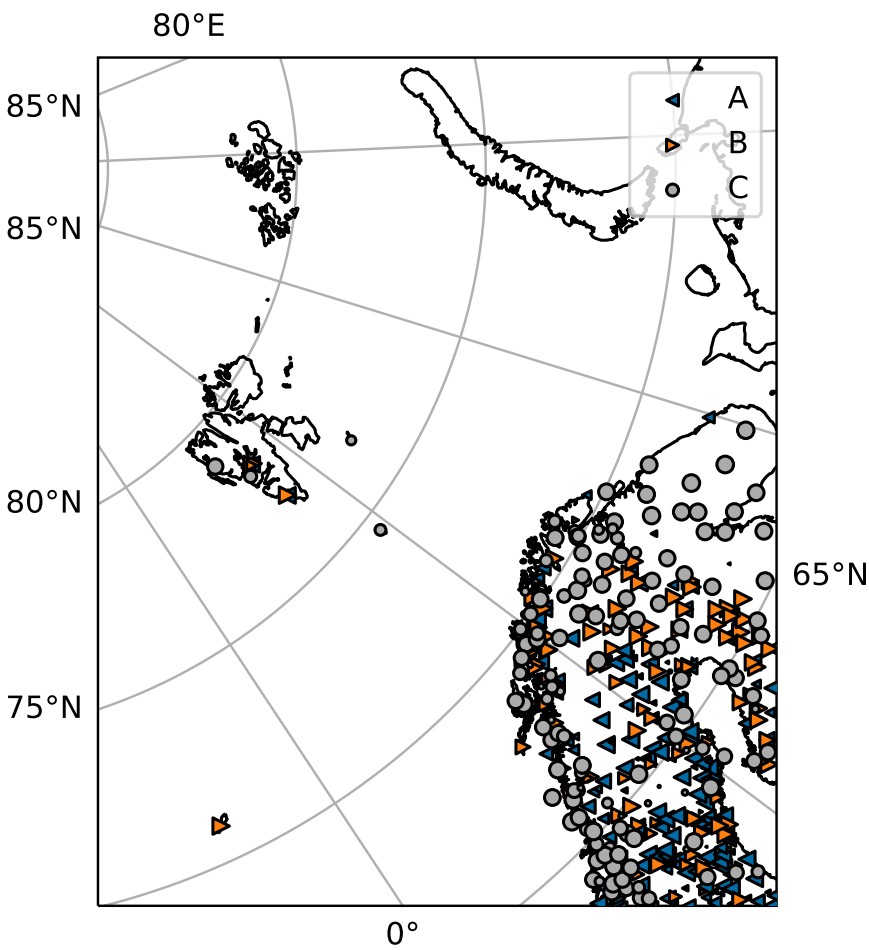

**Figure 2.** Observations of snow depth only used in CARRA-Land-Pv1 (set $A$), only in CARRA (set $B$) and in both (set $C$) over the model domain. Marker size indicates the number of observations through the study period.

## 2.7 Experimental setup

Our experiments are set up on the eastern domain of the CARRA reanalysis, covering northern Scandinavia and Svalbard with 2.5 km grid spacing. The simulations are initialized on 1 September 2015 after a 2 month spin up without data assimilation, and cover the next four consecutive winters. Snow data assimilation is performed daily at 06 UTC using in situ snow depth measurements. The assimilation control vector includes snow water equivalent, snow density and snow heat. Snow albedo and snow age are not included since we focus on snow depth observations in this study. However, albedo and snow age might change after the analysis if snow is removed entirely or added on snow free grid points. In line with Yin et al. (2015); Brangers et al. (2024) and references therein, the number of ensemble members was chosen after comparing shorter experiments with different ensemble sizes. Compared to using 20 members, a 10-member ensemble gave no considerable degradation in performance, we therefore opted for a 10 member ensemble. Additionally, an unperturbed control member was run without data assimilation for evaluation purposes. Hereafter, we refer to the unperturbed run as CTRL and the analysed ensemble simulation as CARRA-Land-Pv1 (Prototype version 1). The purpose of CARRA-Land-Pv1 is to provide a more realistic description of the snowpack covering the region.

**Table 4.** Data assimilation and ensemble perturbation parameters

| Parameter | value |
|---|---|
| observation error ($\sigma_o$) | 0.1 m |
| background covariance inflation ($\beta$) | 1 |
| horizontal impact length ($r_h$) | 50 km |
| vertical impact length ($r_v$) | 200 m |
| horizontal length noise | 400 km |
| temporal auto-correlation noise | 1 day |
| spatial consistency remap | 500 km |
| standard deviation remap | 10 km |

## 3 Results

In this section, we present the evaluation of the CARRA-Land-Pv1 dataset covering 1 November 2015 to 1 August 2019. First, in Sect. 3.1 we show diagnostics of the data assimilation system, including average analysis increments and ensemble correlations. Subsequently, we present the evaluation of the resulting snow depth dataset (Sect. 3.2) and validation against in situ observations compared to CARRA and CTRL (Sect. 3.3). Finally, in Sect. 3.4 we present a verification of modelled snow water equivalent versus snow pillow observations.

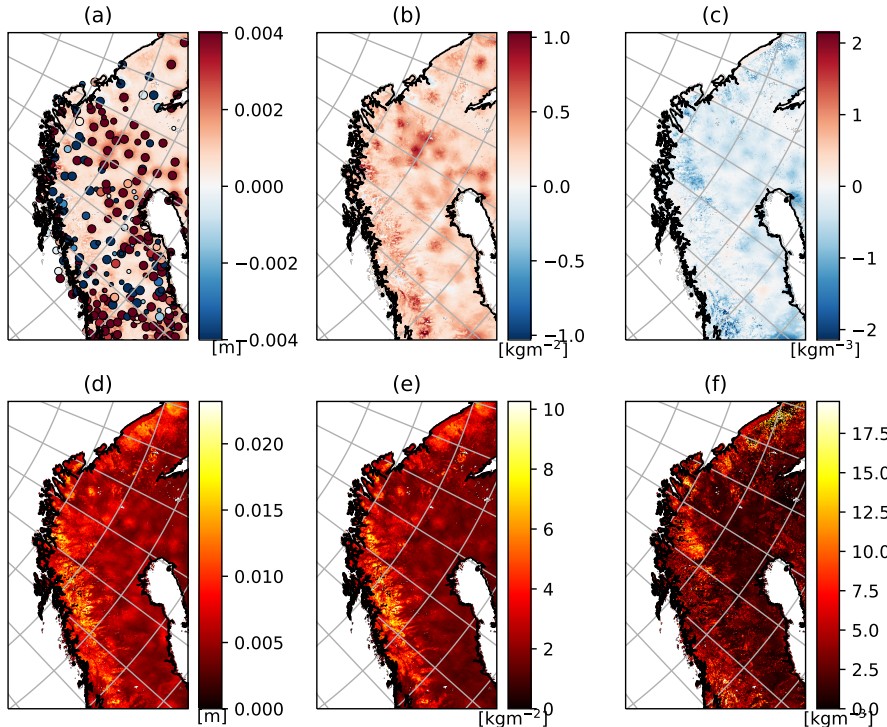

**Figure 3.** Mean/standard deviation (over time) increments of (a/d) total snow depth ($\delta d_{tot}$), (b/e) snow water equivalent ($\delta w_{tot}$) and (c/f) snow density ($\delta \rho_{tot}$). Dots in (a) indicate mean innovation at assimilated observations and their size indicates the number of observations.

### 3.1 Data assimilation diagnostics

We focus on vertically aggregated values, summarizing the values in the 12 layers of the model and define

$$d_{tot} = \sum_i^{12} d_i \qquad w_{tot} = \sum_i^{12} w_i \qquad \rho_{tot} = \frac{w_{tot}}{d_{tot}} \tag{14}$$

for snow depth, snow water equivalent and snow density respectively, where $i$ indicates the model layer of the snow. The
increments presented below are computed from the aggregated variables. For example, the total snow depth increment is
written $\delta d_{tot} = d_{tot}^a - d_{tot}^b$, where $a$ denote analysis and $b$ denote background.

Figure 3 shows maps of the time averaged increments for snow depth $\delta d_{tot}$ (a), snow water equivalent $\delta w_{tot}$ (b), snow density
$\delta \rho_{tot}$ (c) and their respective standard deviations in d-e). Snow depth and snow water equivalent increments are consistently
positive over the domain, while snow density increments are mainly negative. The mean snow depth increments corresponds to
several millimetres each day which roughly accumulates to a metre over one year. For snow water equivalent, about 200 $\mathrm{mm}$
of water is added each year through the increments. The time averaged snow depth innovations $\overline{(y^o - \bar{y}^b)}$ are shown together
with the average snow depth increments (Fig. 3 a). Some observation sites are surrounded by increments with non-zero mean,
particularly in the inland areas. The patterns of these increments are similar to the Gaussian localization function (Eq.10)

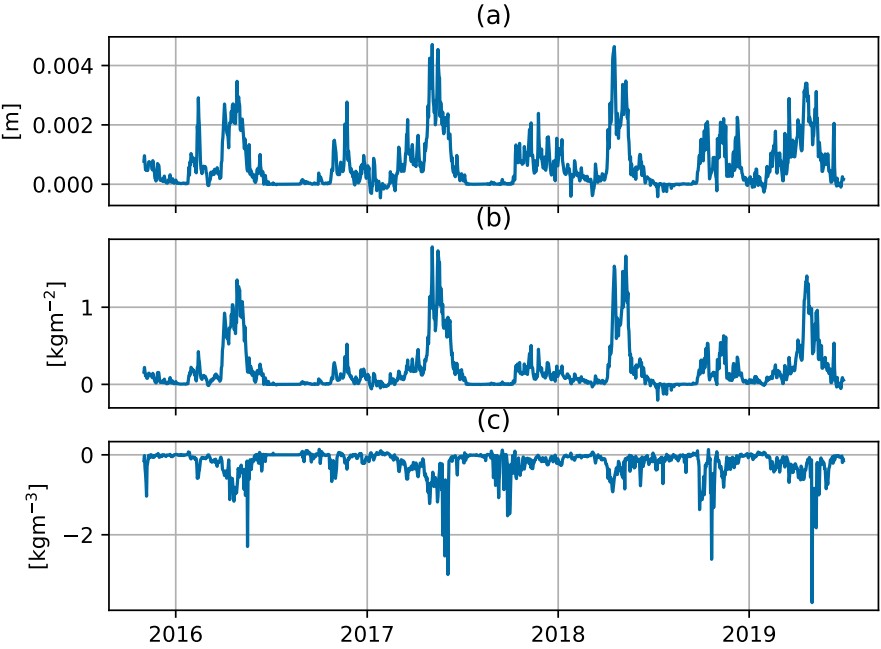

**Figure 4.** Mean (domain average) increments of (a) total snow depth ($\delta d_{tot}$), (b) snow water equivalent $\delta w_{tot}$ and (c) snow density ($\delta \rho_{tot}$).

limiting the impact of the observations and shows the impacted area from each station. The systematic positive increments
are related to positive innovations, indicating too little snow in the model. Similar patterns but with opposite sign are not
seen around stations with negative innovations (indicating too much snow in the model). The stations with mean negative
innovations are situated in mountainous areas and have small impact on the analysis. This is due too the large vertical distance
between model and observation and thus the increment is dampened by the localization (not shown). For snow density, the
stations are not as pronounced as for snow water equivalent and snow depth, and the largest values are found in mountain
areas. The maps of standard deviations indicate larger increment magnitude over mountain areas for all variables, and smaller
over the inland areas. Inland, the pattern around stations seen in mean increments are not as strong, indicating that the areas
suffer from a systematic underestimation. On the other hand, the mountain regions are more exposed to random errors which
cause larger increments, but with less systematic signal. The standard deviations of snow density increments (Fig. 3f) show
fine spatial structure compared to the smoother snow water equivalent (Fig. 3e). This suggests that the topography and thus
temperature forcing might play a bigger role than precipitation for the density uncertainty.

  Time series of domain averaged increments are shown in Fig. 4 for the same variables as Fig. 3. Largest positive snow depth
increments are seen during spring where snow water equivalent increments are positive and density increments are negative.
The early accumulation phase is also dominated by positive snow depth increments, but with smaller density increments.
During the winter, snow depth increments decrease and in some years they become slightly negative before the melting phase

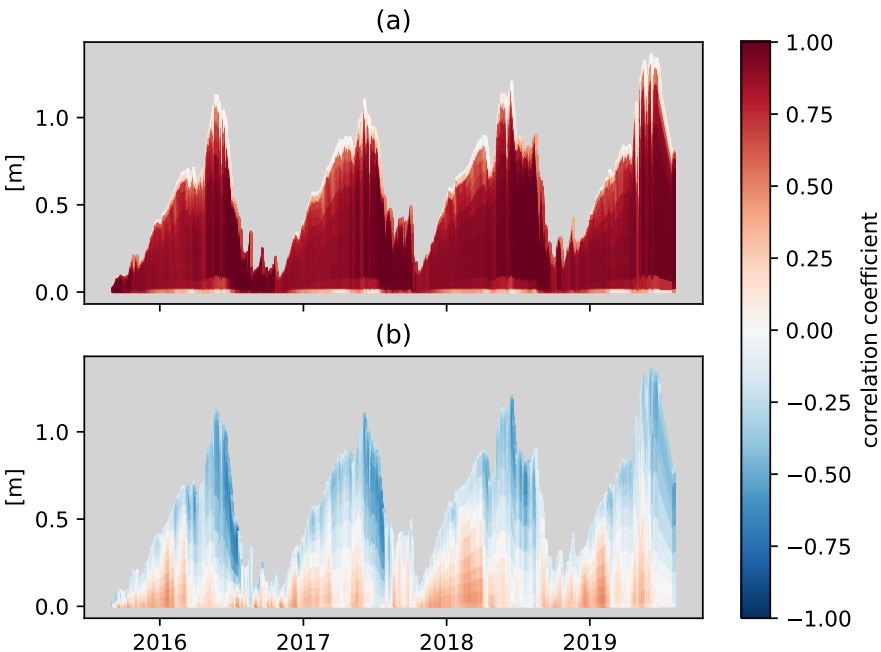

**Figure 5.** Time series of domain average ensemble-mean snow depth $d_{tot}$ with colour-contoured ensemble correlations between total snow depth $d_{tot}$ and snow water equivalent $w_i$ (a) and between $d_{tot}$ and snow density $\rho_i$ (b) per model level using Eq. 12. Only grid points with snow are used, thus the depth is only representative for snow covered grid cells.

begins. The figure suggests that snow is melting (loosing mass and compacting) too rapidly during spring, and that there is a slight underestimation of snowfall in the early accumulation phase.

We continue by investigating the structures of the Kalman gain matrix **K** through the ensemble correlations according to Eq. 12. Figure 5 presents ensemble-mean snow profile time series with ensemble correlations between observation and model variables. Large positive correlations, as seen for snow water equivalent, indicate that a positive innovation (model snow depth is too small) will cause a positive increment to the control variable (increasing snow water equivalent). We note that the upper and lower layers have weak correlations for deep snow, this is a result of the vertical discretization of the snow scheme. The thickness of these boundary layers are limited to resolve the diurnal energy transfer between snow and air and snow and soil. For snow density (Fig. 4b), there is a positive correlation between snow depth and density in the lower layers during accumulation. This can be explained by the compaction of the lower layers in the snowpack due to increased mass in the upper layers during snowfall. The upper layers have negative correlations and strongest in the melting phase. Given a positive innovation in snow depth, the negative correlations would cause the analysis to decrease the density (increase snow depth) to compensate for too high compaction in the model simulation. The introduction of density in the control vector allows the snow depth to be adjusted 1) without changing the snow mass (negative correlations between total snow depth and snow density) and 2) larger change in snow mass compared to only updating mass in the case of positive density correlation.

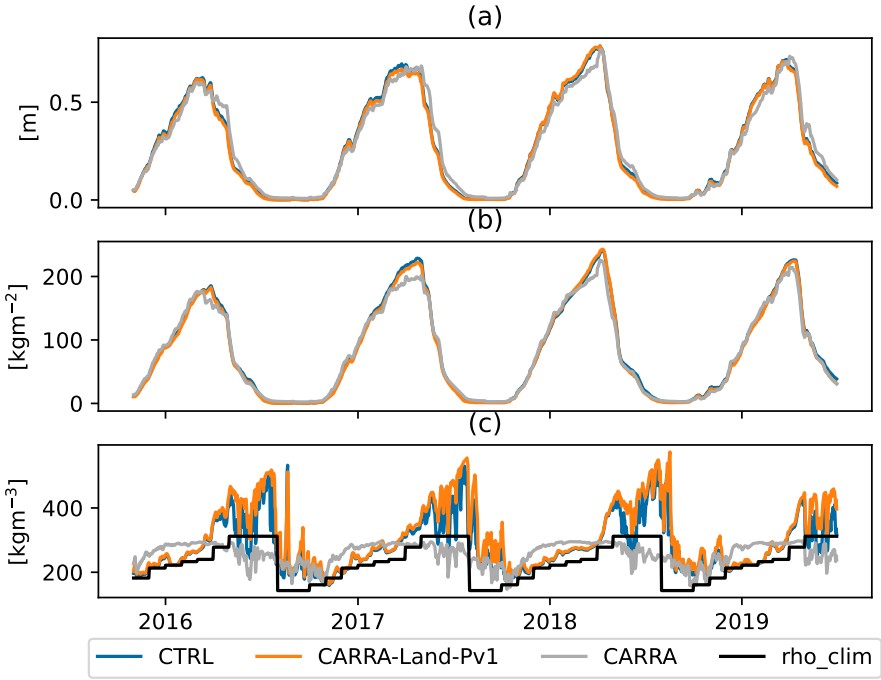

**Figure 6.** Time series of domain average: total snow depth $d_{tot}$ (a), snow water equivalent $w_{tot}$ (b) and snow density $\rho_{tot}$ (c) for CTRL (blue), CARRA-Land-Pv1 (orange) and CARRA (green). Black line indicates climatological snow densities used for converting observations to snow water equivalent in the CARRA system

### 3.2 Seasonal snow characteristics

Figure 6 shows the evolution of domain average snow depth (a), snow water equivalent (b) and snow density (c) for the CARRA-Land-Pv1, CTRL and CARRA. In terms of snow water equivalent, CTRL and CARRA-Land-Pv1 have higher values at the yearly maximum compared to CARRA. In general, there are relatively small differences between CARRA-Land-Pv1 and CTRL. In CTRL and CARRA-Land-Pv1 the evolution of snow water equivalent is smoother compared to CARRA. CARRA tends to have lower snow depth than CARRA-Land-Pv1 and CTRL during the accumulation phase and higher during the melt phase. These characteristics are consistent for all years. The density evolves quite differently in the two snow models (CARRA vs CTRL and CARRA-Land-Pv1). In the single layer snow scheme used in CARRA, the maximum snow density is $300 \mathrm{\,kg\,m^{-3}}$ while the explicit snow scheme (CTRL and CARRA-Land-Pv1) has an upper limit at $973 \mathrm{\,kg\,m^{-3}}$. The snow density reach high values in the melting phase in the multi-layer snow scheme. In CARRA, snow density is increasing faster in the accumulation phase, and stabilises earlier in the winter. During the melting phase the snow density in CARRA decreases compared to the density in the multi-layer models. There is a discrepancy between the CARRA snow density and the climatological values used in its snow analysis. On average, this climatological density is less than the CARRA density before April and greater after May.

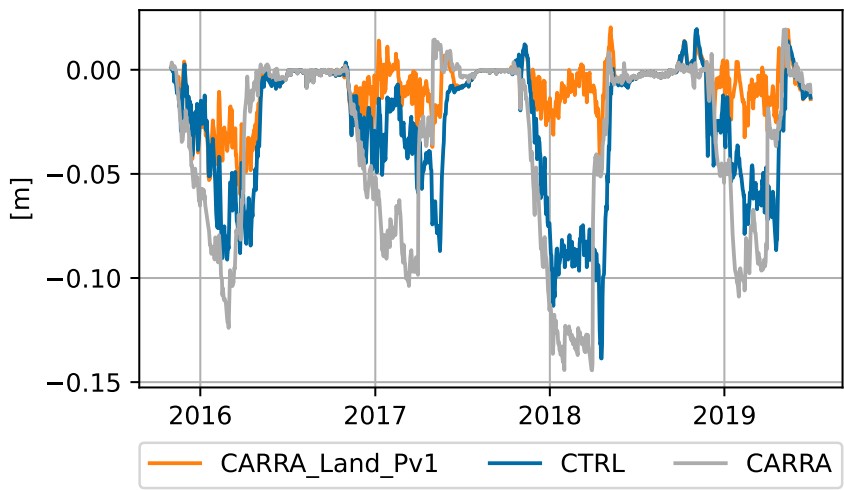

**Figure 7.** Differences between model and observation snow depth $d_{tot}$ averaged over all observation locations.

This coincides with the snow depth differences described above. Comparing CARRA-Land-Pv1 and CTRL shows a slightly increased density in the assimilation experiment.

Time series of mean errors in total snow depth averaged over all observation sites (assimilated and independent) are shown in Fig. 7. All datasets have negative bias in snow depth and the largest bias is seen in CARRA. CARRA-Land-Pv1 has smallest negative bias indicating larger snow depth compared to both CARRA and CTRL.

### 3.3 Evaluation of snow depth

Time series of continuous rank probability score (CRPS) (for ensemble) and mean absolute error (MAE) (for deterministic)
over the three different observational sets are shown in Fig. 8 and Table 3. The CRPS is equal to the MAE for a single member ensemble, or deterministic dataset. Largest errors are seen for CTRL in all sets and yearly maximum varies between 15 and 30 cm. Comparing the three observation sets, the largest errors overall are in OBS-ONLY-CARRA (Fig. 8c). CARRA has larger errors than CARRA-Land-Pv1 in OBS-ONLY-Pv1 and OBS-BOTH (Fig. 8a,b), with values ranging from 10 to 20 cm and 5 to 10 cm respectively. For CARRA-Land-Pv1 a degradation compared to CARRA is seen for OBS-ONLY-CARRA (Fig. 8c) but
it is consistently better than CTRL for all observation sets. We also note a significant drop in errors in CARRA during spring, this is seen for all years. This drop coincides with the transition from March to April and thus new values for climatological density used in CARRAs snow data assimilation From Fig. 6 (c), April is the month where the climatological densities fit best with the model density in CARRA.

We have included both CRPS and MAE of the ensemble-mean to highlight the benefit of CARRA-Land-Pv1 being an
ensemble product. The CRPS values are consistently lower than the MAE of the ensemble-mean. However, relative to the other datasets, the CRPS and MAE of ensemble-mean are quite close. When CRPS for CARRA-Land-Pv1 and MAE of the other

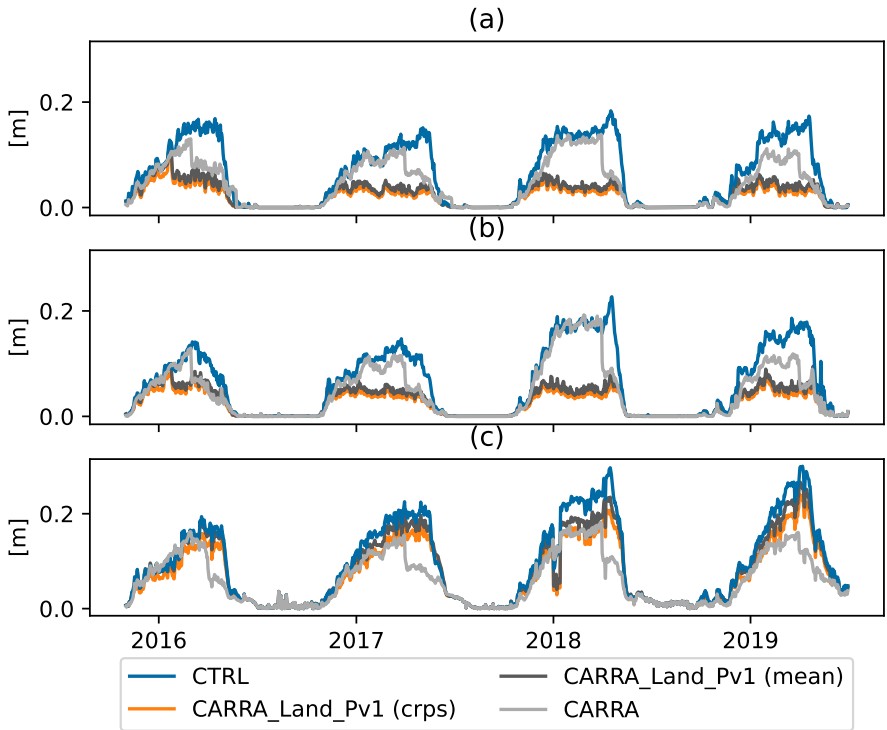

**Figure 8.** Time series of CRPS (for CARRA-Land-Pv1 ensemble) and MAE (for deterministic products) for three different observational sets. (a) shows scores based on OBS-BOTH, (b) OBS-ONLY-Pv1 and (c) OBS-ONLY-CARRA

products are compared, the advantage of using the ensemble is only marginally contributing to the differences. Furthermore, the largest differences between MAE of ensemble-mean and CRPS are seen in the OBS-ONLY-CARRA. This shows how the uncertainty is increased in regions with few observations.

Maps of the difference in CRPS between CARRA-Land-Pv1 and CARRA for the three observation sets are shown in Fig. 9. For the stations in OBS-BOTH (a) and OBS-ONLY-Pv1 (b), CARRA-Land-Pv1 performs better than CARRA/CTRL (CTRL not shown) for most station points. This is also the case for the OBS-ONLY-CARRA set (c), except for a number of stations along the Norwegian mountains and a few scattered inland stations. A closer investigation of the points where CARRA has smaller errors, reveals the following characteristics. 1) The station is isolated, through the localization, from other stations, 2) 380 the point is close to another (assimilated) station with different local conditions and systematic differences. In one case we found up to 50 cm difference in snow depth between observations at neighbouring grid points (not shown). We also note that the spatial distribution of observations in each set differs. In OBS-ONLY-Pv1, more observations are found in the southern part of the domain compared to OBS-ONLY-CARRA.

Summary scores are presented in Table 5. To assess the statistical robustness of the values we use bootstrapping with 1000 385 samples to compute the 95% confidence interval. The intervals did not overlap between the models which indicates statistical

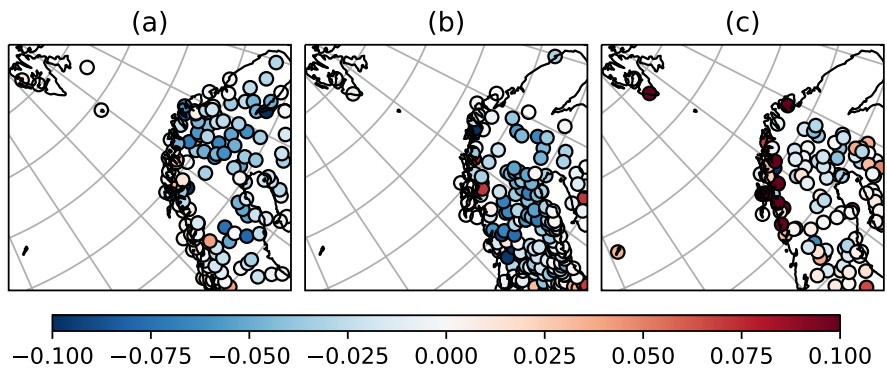

**Figure 9.** Difference in CRPS/MAE scores (CARRA-Land-Pv1 minus CARRA) averaged over time for each observation site. Blue colour indicates smaller errors in CARRA-Land-Pv1 Panels corresponds to observation sets OBS-BOTH (a), OBS-ONLY-Pv1 (b), OBS-ONLY-CARRA (c).

**Table 5.** Summary statistics for the different datasets. Each cell contains values for the three observation sets (OBS-BOTH/OBS-ONLY-Pv1/OBS-ONLY-CARRA). Relative improvements are shown in the two rightmost columns as 100(ref - CARRA-Land-Pv1)/ref

|  | CARRA | CTRL | CARRA-Land-Pv1 | rel CARRA (%) | rel CTRL (%) |
|---|---|---|---|---|---|
| CRPS/MAE | 0.05 / 0.05 / 0.06 | 0.06 / 0.06 / 0.09 | 0.02 / 0.03 / 0.07 | 52.68 / 47.85 / -20.79 | 66.53 / 59.57 / 21.53 |
| Bias | -0.03 / -0.04 / -0.04 | -0.03 / -0.03 / -0.01 | -0.01 / -0.01 / -0.01 | 78.11 / 70.31 / 85.15 | 76.83 / 63.25 / 51.15 |
| Stderr | 0.08 / 0.08 / 0.14 | 0.12 / 0.11 / 0.22 | 0.05 / 0.07 / 0.20 | 32.14 / 15.14 / -46.50 | 55.34 / 42.53 / 6.09 |
| Corr | 0.96 / 0.98 / 0.92 | 0.89 / 0.92 / 0.81 | 0.98 / 0.98 / 0.83 | -2.22 / -0.41 / 9.93 | -9.64 / -5.93 / -2.63 |

robustness of the presented values. Compared to CTRL, CARRA-Land-Pv1 improves on all scores in all observation sets. Relative to CARRA, scores are worse over OBS-ONLY-CARRA, except for bias which is improved by 85%. Over OBS-ONLY-Pv1 and OBS-BOTH, CARRA-Land-Pv1 performs better than the other datasets.

## 3.4 Evaluation of snow water equivalent

In the following comparison, we compute the MAE between CTRL, CARRA and CARRA-Land-Pv1 (ensemble-mean) and the six snow water equivalent observation stations available over the domain. Figure 10 shows differences in MAE (CARRA-Land-Pv1 minus CTRL (a) and CARRA-Land-Pv1 minus CARRA (b)) for each station. The dots indicate that snow water equivalent is improved through snow depth data assimilation in four stations and has neutral impact in two. Compared to CARRA, CARRA-Land-Pv1 has improved snow water equivalent in three, neutral in two and degraded in one station. The
areas surrounding the stations with reduced errors compared to CARRA are associated with larger snow water equivalent values. The degraded station is on the other hand surrounded by negative snow water equivalent differences. In inland areas, the snow water equivalent differences are smaller in magnitude, but mostly positive.

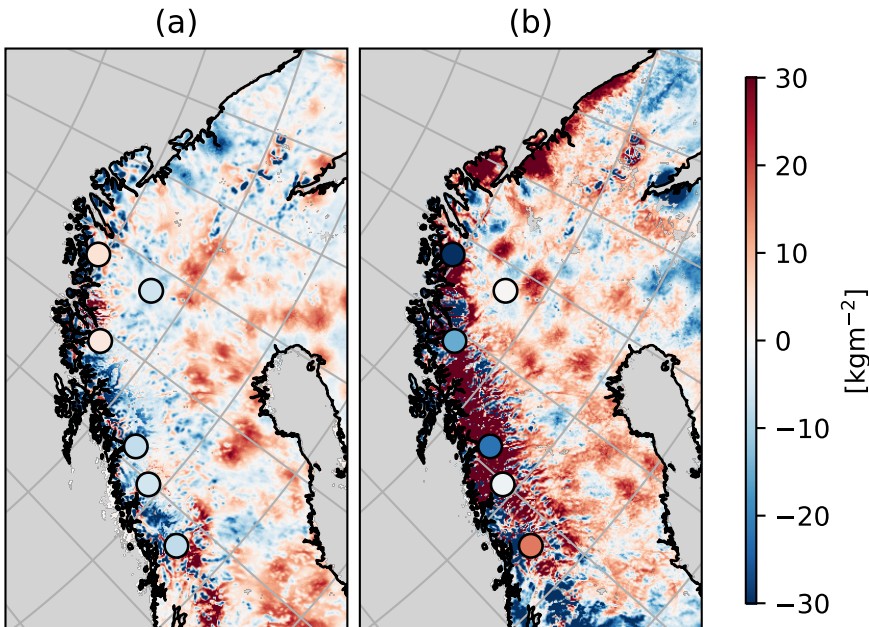

**Figure 10.** Dots representing difference between MAE of CARRA-Land-Pv1 ensemble-mean and references. The references are CTRL (a) and CARRA (b). The MAE is computed from modelled total snow water equivalent $w_{tot}$ and observed snow water equivalent from snow pillow observations. Blue dots indicate smaller absolute errors in CARRA-Land-Pv1 compared to the reference experiment. Colour-shading represents the mean difference in snow water equivalent between the experiments.

## 4 Discussion

Our evaluation of the new regional land reanalysis system (CARRA-Land-Pv1) shows promising results in terms of errors of snow depth estimates compared to CTRL also over independent observations not included in the assimilation. Compared to CARRA, scores are improved in locations where both systems use the observation. However, CARRA-Land-Pv1 is not able to outperform CARRA in locations where only CARRA assimilates the observation. The multi-layer snow scheme also shows the ability to accumulate more mass within the snowpack during the winter, with about 10% more than CARRA at maximum snow depth. Additionally, the data assimilation scheme adds substantial amounts of snow compared to CTRL. The multi-layer scheme allows for much higher snow density during the melting phase, however, this is slightly moderated by the data assimilation increments (negative increments in density). Systematic increments of snow mass over inland areas indicate a negative bias in the precipitation forcing or to rapid melting of the model snow. The snow density increments are smaller over these areas, thus smaller uncertainty of snow density, suggests that precipitation amount is the dominant cause of the underestimation. Over mountain areas systematic increments are smaller for snow mass but higher for snow density. The steep topography leads to larger uncertainties in the forcing temperature, which again impacts precipitation phase and melting processes. The time series of mean analysis increments (Fig. 4) indicates systematic errors during spring. The snow depth

errors in CTRL are also largest during this period. This suggests a limitation in the model parametrization for melting, or an underestimation of precipitation in the CARRA forcing. A similar underestimation of snow depth is also seen in the CARRA dataset, strengthening the latter suspicion. While these results are not sufficient to conclude about potential forcing biases, they demonstrate how the ensemble based analysis method is able to account for different, flow-dependent, situations and regions. Significant improvements due to model configuration was obtained by Monteiro et al. (2024) related to the melting season. Adopting all their settings will be considered in future work.

Investigation of the ensemble correlations (Fig. 5), indirectly by the Kalman gain matrix, gives insight in how the snow depth observations are used to update the multivariate control vector. The correlation between total snow depth and snow water equivalent is close to 1, stating a strong relationship. This indicates that the common approach in the literature to only update snow water equivalent in snow data assimilation systems is efficient and reasonable, also for multi-layer schemes. By keeping the density constant, snow water equivalent can easily be adjusted to correct the total snow depth. However, the ensemble correlation between total snow depth and snow density (Fig. 5b)) indicates important contributions from snow density. Two main processes are represented by the correlations: 1) compaction during accumulation and 2) compaction due to melting. These correspond to positive and negative correlations between total snow depth and snow density respectively. The first in lower layers and the latter in the upper layers. These results highlight the benefit of including snow density in the data assimilation even if only total snow depth observations are used. Neglecting the error in snow density, by keeping it constant in the assimilation, could lead to errors in snow mass both during the accumulation and the melting phase. During the accumulation, when the increased snow depth is related to increased snow density, increments are on average close to zero which indicates less systematic difference between snow depth and observations. The increments of snow density are on average largest in magnitude and negative during the melting phase (Fig. 4c)). This indicates positive innovations (model snow depth too low) during melting. If only the snow water equivalent was updated, more snow mass would have to be added in order to adjust the snow depth towards the observation. During melting season, this would impact the amount of run-off and potentially downstream use. According to Günther et al. (2019), input data is the most important source of snowpack uncertainty. However, model structure and parameters had larger contributions during the melting season. In this study we only perturb input data, the ensemble correlations are thus only a result of the forcing data and relationships of opposite sign could occur if model parameter uncertainty was considered.

Unfortunately, collocated observations of snow depth, snow water equivalent and snow density are (to our knowledge) not available in our study area. The few snow water equivalent observations show mixed results, but indicate an overall positive impact of the system. Since snow depth is generally improved, the uncertainty of snow water equivalent lies mainly in the snow density. Our investigation of ensemble correlations suggests that the assimilation scheme is able to produce reasonable corrections also to density. In studies where more snow water equivalent measurements for validation was available, comparable ensemble-based data assimilation systems gave improved snow water equivalent estimates (Magnusson et al., 2017; Smyth et al., 2020).

By exploring the different observation sets used in CARRA and CARRA-Land-Pv1, we evaluate the analysis performance at grid points where observations are not available for assimilation. Despite that CARRA-Land-Pv1 performs worse than CARRA

over OBS-ONLY-CARRA, errors are consistently reduced compared to CTRL for these stations. The errors of CARRA are also relatively large over the same observation sites in OBS-ONLY-CARRA. These findings are in line with results in Oberrauch et al. (2024). Nevertheless, the distance based localization used in CARRA-Land-Pv1 cause several stations in OBS-ONLY-CARRA to be unreachable in the analysis causing no impact of the assimilation. The degradation compared to CTRL thus come from the perturbed forcing.

Comparison studies should assimilate an identical set of observations in both systems and reserve a fully independent dataset for evaluation. Such a setup would allow a more rigorous assessment of the spatial performance of the two data assimilation methods (CARRA vs CARRA-Land-Pv1). While this was not feasible in the present study due to practical constraints, we highlight it as an important recommendation for future work. The comparison between CARRA and CARRA-Land-Pv1 in this study is thus non-conclusive and should be read with this is mind. However, the comparison between CARRA-Land-Pv1 and CTRL is rigour and is not limited by the experiment configuration.

Cluzet et al. (2022) evaluated different localization strategies for in situ snow depth measurements in mountain regions and found that using ensemble correlation between variables at different grid points rather than distance to localize the analysis was beneficial. Such method should also be evaluated for our system and study area. Although smaller improvements are seen over mountain areas for CARRA-Land-Pv1 comparing to in situ observations, the topography has strong impact on precipitation. This suggests that also the uncertainties should be modified in these areas. We account for topography in the precipitation remapping, but not in the spatial structure of the other noise fields perturbing the amplitudes. We thus suggest that more work is needed to improve the quality of spatial structures for the forcing ensemble.

In this study we used the model grid average snow depth as observation equivalent. However, we found cases where observations close to each other had very different snow depths, which indicates large sub-grid variability. There are also potentially systematic differences in snow conditions between the low and high vegetation patches, and, likely, the observation is not representative of the grid average value. This deficiency could also explain the systematic increments during melting phase as discussed above. For example, in a grid cell represented by 50% low and 50% high vegetation, a snow depth observation reads 40 $\mathrm{cm}$. Furthermore, in the high vegetation (forest) patch of the model, the snow is gone, while in the low vegetation patch (where most snow observations are located) the model has 50 $\mathrm{cm}$ of snow. Using the grid average observation operator will give an observation equivalent of 25 $\mathrm{cm}$ resulting in an innovation of 15 $\mathrm{cm}$. Consequently, this will typically cause snow to be added. However, given that the observation is representative of an open area, we could use the low vegetation patch only to compute the observation equivalent which results in a negative innovation of -10 $\mathrm{cm}$, and snow will be removed. Another important case is areas where snow is redistributed by wind (Gisnås et al., 2016; Aas et al., 2017). This process is not represented in the ISBA explicit snow model, however if accounted for in the observation operator the observations can still be used to correct the model values. In emission modelling, statistical models for subgrid processes have been used to counter similar problems (Koohkan and Bocquet, 2012).

A fundamental difference between CARRA and CARRA-Land-Pv1 systems is the flow dependency of the ensemble-based CARRA-Land-Pv1 versus the static CARRA assimilation scheme. Where the CARRA system will treat any value of an observation equally, but CARRA-Land-Pv1 will account for the background uncertainties. For instance, if all members are snow-

free, no observation of snow depth results in non-zero increments. This has potential weaknesses, however, as long as the ensemble has reliable uncertainties it can prevent unwanted increments due to erroneous observations. Although not presented, we found several cases of inconsistencies within the CARRA dataset. On the other hand, the CARRA-Land-Pv1 shows more
consistent time series and spatial patterns. Due to more confidence in the model background, single observations have less impact. However, we have seen that small increments over time ensures that the estimates follow observations closely.

Since the data assimilation method is ensemble based, the uncertainty of the output variables can also be made available for the user. We found that the CRPS was marginally smaller than the MAE of the ensemble-mean, but larger differences (more benefit of the ensemble) was seen in the OBS-ONLY-CARRA observation locations Fig. 8 c). This demonstrates that where
less or no observations are available the uncertainty increases. For the areas with good observation coverage, the uncertainty should be smaller, thus a narrower ensemble spread.

## 5    Conclusions

In this study, we implement a system for regional land reanalysis with enhanced representation of snow processes through a multi-layer snow scheme and the ensemble-based LETKF for data assimilation. A 4-year dataset is produced with the system
assimilating in situ snow depth observations. The dataset is evaluated and compared to existing products covering the European Arctic. The snow depth estimates show reduced errors compared to the reference datasets and consistent positive impact of the data assimilation. However, limited impact is found in mountain regions along the Norwegian coast, which highlights fundamental challenges of snow modelling and assimilation in these regions.

Through perturbation of forcing data, the ensemble is able to represent uncertainty related to compaction processes. This
allows the data assimilation to make corrections to the state variables accounting for these processes which are usually neglected when only snow water equivalent is updated. Furthermore, the method presented demonstrates how season dependent processes can be accounted for in the assimilation which is not possible with the reference OI method.

Systematic positive snow depth increments during spring are discussed and model parametrization and precipitation bias in the forcing data are highlighted as possible explanations. However, both of these deficiencies are corrected by the assimilation.
Conversely, the representativeness errors of the observations are found to be large and, in some cases, impact the assimilation performance negatively. We thus suggest future studies to develop more advanced observation operators for in situ snow measurements accounting for sub grid conditions (e.g. following Koohkan and Bocquet, 2012).

For seasonal stream flow predictions, accurate estimates of snow water equivalent are crucial (Casson et al., 2018). The multi-layer surface scheme shows up to 10% difference in accumulated snow water during the maximum snow depth compared to
CARRA. Moreover, the dataset shows less jumpy time series of snow water equivalent and its errors. This suggests that there is a potential impact on downstream usage of this product, with more reliable estimates of snow water equivalent through the snow season.

While the assimilation of in situ snow depth observations gives an overall positive impact, the spatial coverage is a limitation of these observations. This is seen through the comparison with CARRA over the OBS-ONLY-CARRA observations.

To provide the best snow estimates covering all parts of the domain, more observations need to be assimilated. Satellite based instruments operating in both visual and microwave frequencies have shown to provide accurate estimates of different snow quantities (De Lannoy et al., 2012; de Rosnay et al., 2014; Charrois et al., 2016; Micheletty et al., 2022; Gichamo and Draper, 2022). With relatively frequent revisit times at high latitudes, these are attractive sources of information. Including new types of observations requires the implementation of an observation operator. For satellite observations, this can either be radiative transfer models like the Snow Microwave Radiative Transfer (SMRT) (Picard et al., 2018) or machine learning based observation operators (Kwon et al., 2019).

The system presented is also flexible in terms of extending the analyses to more than snow state. Herbert et al. (2024) demonstrated the benefit of unified surface data assimilation. The system used in this work is well suited for combining soil and snow data assimilation in a unified approach.

*Data availability.* Snow depth observations used in this study are obtained from the Meteorological Archival and Retrieval System (MARS) of the European Centre for Medium-Range Weather Forecasts (ECMWF) (restricted access). The Snow pillow observations were fetched from the Hydrological API (HydAPI) provided by the Norwegian Water Resources and Energy Directorate (NVE) hydapi.nve.no. CARRA data is available from the Climate Data Store (CDS) (Schyberg H. et al., 2021). The CARRA-Land-Pv1 dataset is available from the corresponding author upon reasonable request.

## Appendix A: Appendix

In the following, the remapping procedure is described step by step.

1. Construct a spatial correlated vector field $\mathbf{v}$. In this study, a 2D convolution with a Gaussian kernel function was used to obtain spatial correlations of a 2D random variable. This controls the spatial consistency of the magnitude and direction of the advection. To constrain displacement of precipitation over mountainous regions, the vector field is multiplied with a scalar field $\gamma$ given:

$$\gamma = \frac{1}{1 + \exp\left(15(\tilde{\gamma} - 0.3)\right)} \tag{A1}$$

where $\tilde{\gamma}$ is a normalized quantity derived from the smoothed product of surface elevation and slope magnitude:

$$\tilde{\gamma} = \frac{\gamma_0 - \min(\gamma_0)}{\max(\gamma_0) - \min(\gamma_0)} \tag{A2}$$

$$\gamma_0 = K * (\|\nabla z_s\| z_s) \tag{A3}$$

Here: $z_s$ is the surface elevation field, $\|\nabla z_s\|$ is the gradient magnitude of the elevation, $K$ is a Gaussian kernel used for spatial smoothing via convolution, $\tilde{\gamma} \in [0, 1]$ after normalization. The sigmoid function applied to $\tilde{\gamma}$ sharpens the transition around the threshold value 0.3, controlling the influence of terrain features.

2. Let the vector $\mathbf{I}$ be the grid indices of the domain, mapping $P \to P$. Here $P$ is the field to be remapped.

3. Advect $\mathbf{I}$ by $\mathbf{v}$ to obtain $\mathbf{I}'$

$$\mathbf{I}' = \mathbf{I} - \mathbf{v} \cdot \nabla I \tag{A4}$$

4. Compute $P'$ by bilinear interpolation of $P(\mathbf{I}')$

$$P'_{i,j} = (1-\alpha)[\beta P_{i^-{}',j^+{}'} + (1-\beta)P_{i^-{}',j^-{}'}] \qquad\qquad + \alpha[\beta P_{i^+{}',j^+{}'} + (1-\beta)P_{i^+{}',j^-{}'}], \tag{A5}$$

where $\alpha = i' - i^-{}'$, $\beta = j' - j^-{}'$, the superscripts $-\prime$ and $+\prime$ indicate the advected index $i'$ rounded down and up to nearest integer respectively.

Since precipitation has moved within the domain, the phase might have changed and a redistribution of snow and rain is necessary. If a grid cell have precipitation before the remapping, the original ratio $r$ between snow and total precipitation is used. However, if it does not exist (total precipitation was zero), the forcing temperature $T$ is used as

$$r = \begin{cases} 1 \text{ if } T \leq 272 \\ 1 - \frac{T-272}{275-272} \text{ if } 272 < T < 275 \\ 0 \text{ if } T \geq 275 \end{cases} \tag{A6}$$

*Author contributions.* Experimental design, setup and simulations: ÅB and JB, analysis, visualization and draft writing: ÅB, review and editing: JB and MM.

*Competing interests.* The authors declare that they have no conflict of interest.

*Acknowledgements.* We would like to thank Matthieu Lafaysse, Nima Zafarmomen and an anonymous reviewer for valuable feedback. We believe their inputs significantly improved the quality of the article. The work in this paper was conducted as part of The Copernicus Climate Change Service Evolution (CERISE) project (grant agreement No101082139), funded by the European Union. Views and opinions expressed are however those of the author(s) only and do not necessarily reflect those of the European Union or the Commission. Neither the European Union nor the granting authority can be held responsible for them.

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
