# Peer review of "Ensemble-based snow depth data assimilation for a multi-layer snow scheme over the European Arctic"

_EGUsphere, 2025_

## Referee Comment (RC2)

Review of
*Ensemble-based snow depth data assimilation for a multi-layer snow scheme over the European Arctic*
by Åsmund Bakketun, Jostein Blyverket, and Malte Müller

**General comments**

This manuscript presents a regional reanalysis system for snow water equivalent. It is named CARRA-Land-Pv1 and is based on a multi-layer snow model (ISBA, with 12 vertical layers) and a Local Ensemble Transform Kalman Filter to assimilate snow depth in the model. This new reanalysis is evaluated against a reference analysis, CARRA, which also used ISBA but as part of a larger model that also includes all the other components of a numerical weather prediction model. Therefore, CARRA and the new CARRA-Land-Pv1 proposed by the authors share the same snow model and the same meteorological forcings (generated by CARRA) but they differ in 1) the assimilation method itself and 2) the observation sets that are assimilated (partially, as some observation stations are used by both systems). The two systems are also compared against observations from six completely independent snow pillow stations that are not assimilated in CARRA nor CARRA-Land-Pv1. The authors show that, in general, their proposed new reanalysis outperforms CARRA.

I believe the topic is of great interest to the readers of The Cryosphere. However, there are a few methodological aspects that need to be clarified, and I also found minor linguistic issues that need to be corrected. Specifically, details regarding the perturbation of meteorological forcings need to be added, and the LENKF method needs to be explained better. Finally, I could not find the resolution of the grid anywhere. I apologise if I have missed it, but if it is not mentioned, it should.

**Specific comments**

1. Introduction: too much emphasis on mountains
2. Line 118: The terms in equation (1) are not adequately defined, as it is not indicated what « x » stands for. You should take what is written at line 159 (« where x represents the ensemble control vector (…) a and b indicate analysis and background (…) » and place it on line 118 instead.
3. Lines 143-148: You mention on several occasions that the state vector has high dimensionality. It would be good to provide the reader with numbers regarding what size is small, what size is average, and what size is large. In addition, I think a more detailed explanation of the concept of localization is needed. In particular, how do you ensure that neighboring points remain correlated (preserving the spatial structure of snow depth and SWE) if data assimilation is performed independently point by point?
4. Figure 1: I cannot understand this figure. I have read the text several times, I still don't understand the remapping. I understand that the purpose is to better represent the spatial uncertainty of precipitation, but I don't understand the method and Figure 1. Is it possible to modify Figure 1 to make it clearer? Maybe provide an example with numbers and/or real grid cells? The Appendix also did not help.
5. Section 2.4: Please provide a table that indicates the range of perturbations used for each variable, as well as any relevant equations.
6. Line 310: You find the ensemble to be only marginally better than the average of all its members. However, I'm wondering about the potential users of that new reanalyse. Do they want only deterministic estimates of snow depth, or is there value for them in having access to the full ensemble and information about the uncertainty? I think this should be discussed.
7. Line 316: I'm not sure « climatology » is the right word for geographically close stations. I understand that they might have differences in their observation records, but can we really talk about different climatology?

**Linguistic comments**

8. Abstract, lines 2-4: « merge information from the two sources » is not clear, as the « two sources » could be understood as either « prediction systems » and « historical reanalysis » or « observations » and « physical laws in models ». The reader has to logically deduce that you are referring to the latter, but it is not clear from the way the sentence is written. I suggest reformulating.
9. Abstract line 11: Is it possible to replace « relatively large » by something more precise?
10. Line 28, remove the comma between « models » and « is »
11. Line 155: This group of equations should be numbered
12. Line 167, there is an « s » missing in « represents»
13. Lines 382-383:  The sentence « In emission modelling, the use of statistical models for subgrid processes has been used to encounter similar problems » Needs to be reformulated. I suggest « In emission modelling, **statistical models for subgrid processes have been used** to **counter** similar problems
14. Line 325: the word « stations » is missing between « observation » and « available »

---

## Author Comment (AC1)

**Response letter to RC2**

**Dear RC2,**

We thank you for the review comments, we believe that the manuscript is strengthened after addressing them. In this letter the original comments are found with our response in red color.

**General comments**

This manuscript presents a regional reanalysis system for snow water equivalent. It is named CARRA-Land-Pv1 and is based on a multi-layer snow model (ISBA, with 12 vertical layers) and a Local Ensemble Transform Kalman Filter to assimilate snow depth in the model. This new reanalysis is evaluated against a reference analysis, CARRA, which also used ISBA but as part of a larger model that also includes all the other components of a numerical weather prediction model. Therefore, CARRA and the new CARRA-Land-Pv1 proposed by the authors share the same snow model and the same meteorological forcings (generated by CARRA) but they differ in 1) the assimilation method itself and 2) the observation sets that are assimilated (partially, as some observation stations are used by both systems). The two systems are also compared against observations from six completely independent snow pillow stations that are not assimilated in CARRA nor CARRA-Land-Pv1. The authors show that, in general, their proposed new reanalysis outperforms CARRA.

I believe the topic is of great interest to the readers of The Cryosphere. However, there are a few methodological aspects that need to be clarified, and I also found minor linguistic issues that need to be corrected. Specifically, details regarding the perturbation of meteorological forcings need to be added, and the LENKF method needs to be explained better. Finally, I could not find the resolution of the grid anywhere. I apologise if I have missed it, but if it is not mentioned, it should.

Indeed the grid resolution was missing in the original manuscript. We add this in the revised version. Note also that the configuration of ISBA is different in CARRA (single layer) and CARRA\_Land\_Pv1 (multi-layer).

**Specific comments**

1. Introduction: too much emphasis on mountains

We think that mountains are particularly important in terms of seasonal snow as they receive more snow than lower-laying regions and are subject to larger uncertainties in snow estimation products.

2. Line 118: The terms in equation (1) are not adequately defined, as it is not indicated what  $(x \times x)$  stands for. You should take what is written at line 159 ( $(x \times x)$ ) where  $(x \times x)$  represents the

ensemble control vector (...) a and b indicate analysis and background (...) » and place it on line 118 instead.

We thank the reviewer for pointing out this missing detail, we add definitions of  $x^a$  and  $x^b$  (these are not necessary ensembles at this point).

3. Lines 143-148: You mention on several occasions that the state vector has high dimensionality. It would be good to provide the reader with numbers regarding what size is small, what size is average, and what size is large. In addition, I think a more detailed explanation of the concept of localization is needed. In particular, how do you ensure that neighboring points remain correlated (preserving the spatial structure of snow depth and SWE) if data assimilation is performed independently point by point?

We agree that the wording is vague, we include an example of a typical number and how it is reduced using the local filter. We also improve the description of the LETKF and how information is distributed spatially.

4. Figure 1: I cannot understand this figure. I have read the text several times, I still don't understand the remapping. I understand that the purpose is to better represent the spatial uncertainty of precipitation, but I don't understand the method and Figure 1. Is it possible to modify Figure 1 to make it clearer? Maybe provide an example with numbers and/or real grid cells? The Appendix also did not help.

We understand that Figure 1 could be difficult to interpret, we will thus suggest the following figures to replace it.

Fig 1R a) Precipitation field on model resolution with a small (3 member) ensemble. Upper left panel shows precipitation rate from the CARRA dataset, upper right and lower panels

show ensemble realizations after remapping is applied. The star marks a reference point, for example an observation site.

Fig 1R b) Colors indicate ensemble correlations between the values in the point marked by star and at the respective pixel. Red colors indicate positive correlations. If more snow than the model was measured at the marker, snow would be increased/decreased at the red/blue areas. Note that localization is not applied in this figure.

We hope these figures are more intuitive as they represent model derived precipitation. We also clarify the text describing the method and extend the appendix with more details.

5. Section 2.4: Please provide a table that indicates the range of perturbations used for each variable, as well as any relevant equations.

We include perturbation parameters and extend the description of the perturbation methods.

6. Line 310: You find the ensemble to be only marginally better than the average of all its members. However, I'm wondering about the potential users of that new reanalyse. Do they want only deterministic estimates of snow depth, or is there value for them in having access to the full ensemble and information about the uncertainty? I think this should be discussed.

We thank the reviewer for this comment, and believe this point is not covered sufficiently. We extend the discussion to include the aspect of ensemble uncertainty. Specifically we discuss the fact that the CRPS is better relative to the MAE of the ensemble mean for the OBS-ONLY-CARRA observations. This suggests that the ensemble is quite narrow and depends more on observation error than background error. On the other hand, away from observations, the ensemble has more spread indicating a less certain estimate.

7. Line 316: I'm not sure « climatology » is the right word for geographically close stations. I understand that they might have differences in their observation records, but can we really talk about different climatology?

We replaced "climatology" with "local conditions and systematic differences"

**Linguistic comments**

8. Abstract, lines 2-4: « merge information from the two sources » is not clear, as the « two sources » could be understood as either « prediction systems » and « historical reanalysis » or « observations » and « physical laws in models ». The reader has to logically deduce that you are referring to the latter, but it is not clear from the way the sentence is written. I suggest reformulating.

We replace "information from the two sources " with "information from model estimates and observations" to clarify.

9. Abstract line 11: Is it possible to replace « relatively large » by something more precise?

We agree that "relatively large" is not precise and believe "multivariate" is a better word here.

10. Line 28, remove the comma between « models » and « is »

**Done**

11. Line 155: This group of equations should be numbered

**Done**

12. Line 167, there is an « s » missing in « represents»

**Done**

13. Lines 382-383: The sentence « In emission modelling, the use of statistical models for subgrid processes has been used to encounter similar problems » Needs to be reformulated. I suggest « In emission modelling, statistical models for subgrid processes have been used to counter similar problems

We adopt the suggested sentence.

14. Line 325: the word « stations » is missing between « observation » and « available »

Add "stations"

---

## Author Comment (AC2)

**Response letter to RC1**

Dear Matthieu Lafaysse,

We thank you for a thorough review of our manuscript. We have done our best to address your comments and believe that this has strengthened our manuscript. Below follows the reviewer comments and our response in red colored text.

**General comments**

Bakketun et al. present the results of a new land surface reanalysis refining the snow scheme compared to the forcing CARRA reanalysis and focusing on the assimilation of in-situ snow depths with a new ensemble assimilation framework derivated from the well-known Ensemble Kalman Filter. The introduction presents clearly the context and objectives of this work and the production of an Arctic land surface reanalysis with improved snow modelling and snow data assimilation has certainly a great potential for various related scientific fields.

However, in general, the methods are not described with a sufficient accuracy level to allow their reproductibility (parameters used for perturbation, interpolation, observation errors, inflation, etc.). I believe the manuscript shoud be revised in that spirit. As a result, it is sometimes difficult to evaluate the relevance of the proposed methodology, see in particular my detailed comments relative to L149-151, L153-163; L214-217; L250. My understanding after reading the Methods section was that the main advantage of the EnKF was lost by the local assumption for application while the estimation of the snow model states are much more challenging to obtain than in the context of particle filters or particle batch smoothers. More explanations are necessary to understand this key methodological choices, because although I may have misunderstood something, at this stage, my feeling was that the chosen methodology might not be the most appropriate for this application. When reaching the results Section, I understand that LETKF is actually more complex than described, probably including neighbour pixels in the assimilation process but the readers should have fully understood that after reading the Methods even for readers not aware of all the variants of ensemble data assimilation algorithms.

Another critical aspect of this manuscript is the fact that the independence of observations is not considered as a mandatory property for evaluating the added value of data assimilation. Although some datasets are more or less independent, the results do not focus on this part. More critically, when considering only this dataset, I am afraid that the results are not really convincing to demonstrate the added value of their land surface reanalysis compared to the CARRA reference. As a result, I stay with a mitigated opinion where I see a really nice attempt to produce snow reanalyses with state-of-the-art methods, but leading to not really convincing results, questioning whether the chosen assimilation methodology was really appropriate, and with an unclear understanding of the behaviour of the assimilation algorithm due to unsufficient methodology description and maybe lack of examples on specific cases.

Therefore, I would recommend a major review for this manuscript to give the opportunity to authors to improve their description of methods, refine their evaluation process focusing on independent dataset, and maybe find new results emphasing better the added value of their methodology compared to their benchmark.

This being said, I am not fully convinced that assimilating only in situ snow depth observations has really a significant potential in mountainous areas with sparse observations with low spatial representativity, and I am really wondering why the authors do not prefer to start by assimilating snow cover fractions already assimilated in CARRA.

We understand from your comments that our methods have not been described with sufficient detail and particularly how the LETKF is able to spread point observations in space. In the revised manuscript, we provide more detailed information to clarify this matter. To address your comment concerning scientific reproducibility we revise the entire methods section, and add tables with perturbation parameters, localization functions and length scales. The appendix A is also extended with more details and formalism.

A concern is raised about the validation data and its independence. While it is true that the only completely independent validation data are the six snow pillow stations, we argue that the semi-independent observation datasets are sufficient to demonstrate the capabilities of our method. First, the OBS-BOTH shows clear and fair improvement of the proposed method at observation locations. Clearly this cannot prove anything about the performance remote from observation sites. A novelty of our study is the flow-dependent assimilation of snow depth observations in the multi-layer ISBA-ES snow scheme, we therefore believe this is an important result, as it shows that the ensemble scheme is able to translate total snow depth observations into snow water equivalent and snow density increments that again improve the snow depth. In our experiments the other observation sets are only independent from one of the assimilation systems; they do not provide a fair comparison between CARRA and CARRA-Land-Pv1. However, when comparing CARRA-Land-Pv1 and CTRL over the OBS-ONLY-CARRA set it does give a fair and truly independent evaluation of the LETKF on remote from observations (as these observations are not assimilated in CARRA-Land-Pv1). As suggested, a "leave one out" experiment should provide an independent observation set. however, this also requires that all other observations are equal (in CARRA and CARRA-Land-Pv1). This would require a new setup and a new simulation, with both model systems which is practically not feasible, and thus is beyond the scope of this study. This being said, we revise the results, discussion and conclusion sections to target these concerns in a more critical way and to put more emphasis on the independence and limitations of the evaluation data.

**Detailed comments**

L52 Note that this reference presents twin experiments (with synthetic observations). To the best of my knowledge, real observations of surface temperature have not yet been assimilated in a snow cover modelling system. I suggest to be more accurate on that point.

We thank the reviewer for pointing out this detail, we clarify this in the revised manuscript.

L54-56 The authors classify the DA algorithms according to their management of uncertainties but the algorithms also differ in their ability to preserve the multivariate consistency of model variables. It would also be useful to emphasize this point which is also a strength for instance of particle filters.

This is indeed an important difference between the methods, we extend the revised manuscript to cover this point.

L59-60 Variable transformation is also a common practice to fit with the Gaussian assumption.

L61 Unclear at this step of the introduction which model refers to « the multi-layer snow model ».

As the description of "the multi-layer snow model" follows, we change to "a multi-layer..." at this point.

L85-86 How was estimated this lookup table and is it variable in space?

To the best of our knowledge the origin of the lookup table is not well documented, and must be considered as a code legacy in our NWP system. It is not variable in space. The values are shown in Figure 6.

L97 and elsewhere: The display of units should be improved (space between kg and m-2)

This is corrected in the revised manuscript.

L88 I guess there is a typo in the unit (kg m-2 and not kg m-3). Can you explain the philosophy behind excluding observations « if the model exceeds 25 kg m-2 »?

We interpret the rule as follows: if the model and the satellite observation agree that it is snow, there is no need to use the observation. Note that the threshold of "no snow" observations to be disregarded is 100 kg m-2 in the model which can be considered as a first guess check where the observation is considered to be unlikely.

L90-93 Does this mean that pseudo-observations and in-situ observations are considered to have the same observation error ?

The observation error is not equal for these two observations, we clarify this in the revised manuscript.

L93-94 Are there other variables to describe the snow cover state in the model? Are they kept at the same value during this process?

The single layer snow scheme used in CARRA also represents albedo and snow density, however, these are kept constant during the assimilation. We adjust the sentence to make this clear in the revised manuscript.

L94-95 Can you explain the reason behind that ? I guess this is because surface temperature can not be updated ?

We understand this choice as a way to give more confidence to the model, and to avoid placing snow on the ground when it is "unlikely" e.g. in the middle of the summer. Together with the above threshold of 25 kg m-2 for using observations of snow, we interpret these as measures to represent some sort of uncertainty in the direct insertion method.

L96 « For modelling the land surface »  $\rightarrow$  I understand that the idea is to produce a complementary offline simulation with an improved surface model compared to the CARRA coupled system, but this could be more explicitly explained in this transition.

We thank the reviewer for pointing this out, we rewrite the transition in the revised manuscript.

L101 « the reduced heat flux between soil and snow » As a coauthor of Monteiro et al, 2024, I know what this is about, but I can imagine than most other readers would need more explanations.

Good point, we improve the paragraph to better explain this detail.

L112 To encourage the reproductibility of the results, could you be more accurate on the spatial and temporal interpolation methods used for the different forcing variables?

We agree that this is not described sufficiently. We add more details about the interpolation methods used for spatial and temporal interpolation in the revised manuscript.

L129-130 Snow albedo and snow age do also need to be updated

We update the manuscript so that all the variables are included here. While we do not include age and albedo in our control vector, they are modified if new snow is added on snow-free ground. This is described in the experimental setup section.

L128-132 Direct insertion of snow depth has been applied in some studies with multilayer snow models by preserving the ratio between layers and other snow properties. The assumptions are strong in that case, but it could be mentioned to emphasize the added value of more advanced ensemble approaches.

In the revised manuscript, we elaborate on this topic and include a reference to Brangers et. al 2024 where a similar approach is used.

L149-151 If EnKF is applied independently at each grid point, how can the increments be propagated in space? This is rather counter-intuitive in the context of assimilation of in-situ snow depths in a spatialized system. Please provide explanations.

We understand that our description of the LETKF was not sufficient in the first edition, we extend the presentation of the scheme in the revised version and specifically how information is spread spatially with the LETKF.

L153-163 Could you mention if vector x includes the 36 prognostic variables described line 129 ? If yes, how do you manage snow age and snow albedo ? What is the implication of computing error covariances with such a large vector ? What guarantees that the resulting analysed vector fills the discretization rules of the snow scheme ? Considering that for a given snow depth, the vertical discretization of ISBA-ES is fixed, I am not sure what is the interest to consider the 12 layer snow depths as independent variables in the state vector. I imagine it helps constraining unobserved density and heat contents. But could you justify this choice and provide more details to help the understanding beyond the mathematical formalism ?

Then, does vector x only include the prognostic variables of one single point? If yes, how do you propagate information in space? I see further from the results, that increments are spatialized but I do not see from this formalism how this happens.

In this context "x" includes the 36 prognostic variables for a particular vegetation patch at a grid point. The additional variables, age and albedo are kept constant during the assimilation and are only modified if snow is added on snow-free ground, or removed entirely (described in section 2.5). There is no guarantee that the analysis fulfills the ISBA-ES discretization, but if they don't, the snow scheme applies the redistribution for the next forecast cycle. Note that all members of the background ensemble obey the discretization and that the analysis is a linear combination of these. Also note that in the LETKF, the error covariances are computed in observation space (Yb^T R-1 Yb) which is of lower dimensionality than the model space. The reason for including all layers in the assimilation is to account for the vertical dependent processes responsible for the deviation from the observed condition.

Based on this referee comment, we add more details in Section 2.3 to better describe the spatialisation capacity of the filter.

L165 This assumption would mean that observations are as representative as the open patch than as the forest patch. However, in a large majority, observations are operated on open areas. Could you comment on that ?

This is a very important question and we discuss this in the manuscript (L370) and recommend future studies to investigate the topic. However, we consider this to be out of scope, and choose the grid point average. We include a justification in the methods and data section in the revised manuscript.

L182-183 This is true but monovariate perturbations may produce unrealistic meteorological conditions with physical inconsistencies between meteorological variables, potentially leading to unrealistic model states.

Yes, ensuring absolute- physical consistency is challenging, both between variables, in time and space. This is why we follow the literature and apply a cross-correlation between variables when generating the noise fields to ensure at least first-order dependencies between the perturbed variables.

L185 As mentioned before, the length of the control vector might be reduced thanks to the bijectivity between total snow depth and layer snow depths.

The depths are not part of the control vector and such reduction would not allow for the adjustment of the density profile within the snowpack. The purpose of this study was also to assess a snow analysis that is model agnostic. We show later that the multivariate control vector is not necessarily a problem due to the strong correlation between the layers.

L187-189 The description of perturbations refers to several references which is not accurate enough for reproductibility. Please provide a table with perturbation methods and parameters (variance, auto-correlation time constants, etc.) for each variable.

To provide the readers with sufficient details we include a table with perturbation parameters in the revised manuscript.

L190-203 This is an interesting idea but even in Appendix A, the perturbation parameters to prescribe are unclear and not provided (I guess at least some horizontal distance defining the statistical properties of the random advection?). Could you think again the description of the method as a basis allowing reproductibility?

**We improve the description of this method both in the method section and in appendix A**

L212 « we initialized zero snow members with the ensemble-mean of members with snow. » It is unclear how is computed the mean. The different members have different layer thicknesses. Do you mean that the properties (density, heat) are averaged for each numerical layer regardless its depth in the snowpack? Is it physically realistic? Would not it be sufficient to extract the values of one single member with low snow depth rather than potentially mixing the properties of thin snowpacks with properties of thick snowpacks?

We thank the reviewer for the suggestion. We agree that a more clever sample than the ensemble mean could be beneficial and using the member with least snow sounds reasonable. The idea behind using the ensemble mean was to minimize the risk of unbalancing the ensemble as a mean member would have a neutral impact on the analysis spread. We justify this in the revised manuscript.

L214-217 If the algorithm provides unrealistic states and crashes happen, I am wondering if applying thresholds is really the appropriate option compared to improvements in the assimilation formalism. To my mind, the main added value of the EnKF compared to particle filters is lost if it is applied independently at each pixel point. At the local scale and to assimilate only snow depths, particle filters would be best suited algorithms to retrieve all variable states without any possibility to obtain unrealistic model states, and they have already been largely applied within SURFEX even with more complex snow schemes. The added value of the proposed approach is unclear for me.

The thresholds applied here are not hard limits to the variables, but only used as indication of problematic states to be replaced with likely states (ensemble mean of healthy members). The LETKF does spread observations spatially. Furthermore, this experiment is developed

towards operational NWP and reanalysis systems which need to consider computational cost and timeliness. The system is also suited to include other variables like soil variables in the control vector for unified multivariate surface data assimilation, which is currently not possible in the OI scheme.

Table 1: I do not think a threshold on temperature is necessary as snow heat, snow density and snow mass are sufficient to diagnose snow temperature. Or do you mean the threshold on temperature is applied to constrain the heat content?

We agree with the reviewer that a threshold on temperature might not be necessary. The temperature limit was added here because the snow model also has a lower limit for snow temperature (50 K), which usually is triggered if the snow state is corrupt. The temperature is indeed a diagnostic and values below 200 are probably never reached. To clarify, if any of the conditions in table 1 are true, the entire snowpack is replaced at that point. This means that if swe is negative, density and heat are also replaced, even if they are inside the valid range. This way, the temperature limit acts as a sanity check that could discover unrealistic relationships between density and heat content.

Section 2.6 Could you provide the elevation distribution of the different observation datasets ? Is a threshold applied on elevation as commonly done in NWP ?

Figure 1R

A limit is not applied on elevation, but the elevation difference between the observation and model grid point is included in the localization function with an impact length of 200 m (we update section 2.3 to describe the localization method). Regarding the elevation distribution for each observation set (A, B and C), we were unable to obtain the elevations for the CARRA dataset. However, we show in the Fig. 1R the model elevations at observation locations. This gives an indication of the elevation distribution in the different observation sets. While all sets include stations elevated between 0 and 800m above sea level, the CARRA-ONLY (B) has fewer stations in lower elevation and thus a relatively larger portion of high stations.

I also understand that optical snow cover maps were not used here while they were assimilated in CARRA. This should be more explicit and justified as it could be questionable that an offline land surface reanalysis assimilates less observations than the original coupled reanalysis.

Indeed, ideally the observations would be identical between CARRA and our experiment. While the satellite snow cover product was tested technically it was left out to limit the complexity of the evaluation. We justify this in the revised manuscript.

L231 How are initialized the simulations? Do you apply any spinup to initialize the soil temperature profiles? What is the assumption for snow depth? I guess some pixels are affected by permanent snow?

The simulation was initialized in summer and ran 2 months before the official start date (1 September), then, the first month was left out of evaluation to spin up the ensemble. We add more details about the initialisation in the section (2.7). We disregard grid points with permanent snow in our study, both from assimilation and evaluation.

L242 Do you use independent observation datasets (not used in the assimilation process) to evaluate the system ?

The snow depth observations only assimilated in CARRA (ONLY-CARRA) and the snow pillow observations are independent from our system.

L250 The fact that Figure 3 shows spatial patterns of increments means that something is definitely missing in Section 2.3 to produce the spatial propragation of increments when applying EnKF at the pixel scale. I guess this is associated with « the Gaussian localization function » mentioned L256 but this should be explained in Section 2.3 (providing methodological details as well as control parameters) as after my first reading I thought only one pixel was considered in vector x. The paper should be self-sufficient to understand everything without reading other papers describing LETKF. This raises numerous questions by the way about the localization radius. Is it constant or spatially / temporally variable? Does it depend on the density of the observation network? Etc.

We thank the reviewer for pointing this out, we agree that this is not well described and update the section accordingly.

L256 « The patterns of these increments are similar to the Gaussian localization function ». Without any knowledge about this localization function, I do not understand this comment.

Thank you for making us aware of this, we extend the data assimilation section to describe the localization weights referenced in this comment.

L259-260 « The stations with mean negative innovations are situated in mountainous areas and have small impact on the analysis. » Why? Do you reduce some localization radius as a function of elevation? Methodological details are missing to understand.

Again we thank the reviewer for the comment, we extend the methods section to describe the localization used.

L261 « domain-averaged ». There are two different domains between Figure 2 and Figure 3. To which domain is applied this average ?

The average over the entire model domain, which is similar to the CARRA east domain. This was not shown in the original version of the manuscript. We modify Fig. 2 to show the complete model domain. Furthermore, we clarify that this is the simulation domain in the text.

Figure 5 presents ensemble correlations between between layer variables and total snow depth. The results are useful to understand the assimilation process. However, it is not explained how this information is represented at a given depth in the snowpack as the different ensemble members have different snow depths and layer thicknesses. Do you choose a member? Or compute median thicknesses? Please provide the details in the text.

As the correlations are proportional for the gain matrix of the ensemble mean, we use the layers of the ensemble mean for this figure. In the revised manuscript, we specify this in the figure caption and in the text.

L267-270 The correlations between total snow depth and mass of each layer could be commented at the light of the vertical discretization rules of the snow scheme, explaining easily the weak correlation between the total snow depth and the mass of the surface and bottom layers.

Yes, this is an important point for understanding the figure. We include this explanation in the revised manuscript.

L284-291 I would have expected that the effect of assimilation on density would have been described here (comparing densities obtained with CARRA-Land-Pv1 and CTRL), as done before with snow depth.

Certainly, we add a description of the density differences between CARRA.Land-Pv1 and CTRL in the revised manuscript.

L292-296 The observations used to compute errors are not described. Are they independent from the dataset used for assimilation ? If not, I think it is unsufficient to evaluate a data assimilation algorithm with the assimilated data themselves. Then, the fact that assimilation deteriorates snow water equivalent is really problematic as the main advantage the authors emphasize in their method is its ability to attribute snow depth errors in both mass and density. Unfortunately, the results suggest that this idea fails. A much more simple algorithm assuming that the simulated snow density is correct may have been a better assumption ?

These errors are based on all observations, both those used in assimilation and those independent. One problem is that the independent observations in CARRA-Land-Pv1 are not independent for CARRA and the other way around. Regarding deterioration of snow water equivalent, we want to remind that the figure does not show the actual observed snow water

equivalent but observed snow depth converted to snow water equivalent using the climatological values used in the CARRA assimilation system. We realize that this might be confusing since we also compare with actual snow water equivalent observations, and suggest removing the panel (Fig. 7b) in the revised manuscript.

L300-301 If I understand well, OBS-ONLY-CARRA is the only relatively independent dataset to evaluate the added value of assimilation. So this is of course more challenging, but the description of results in the whole section should be mainly focused on this dataset, as imrproving snow depth at the assimilated observations is definitely not a proof that the system is valid at large scale. The word 'validation' is inappropriate in the title of the subsection (prefer evaluation). Unfortunately, the fact that the whole method deteriorates scores compared to CARRA (Fig 8c, value on the right of each column in Tab 3) is rather problematic while the goal of the study was to improve the simulation of snow cover with a dedicated offline reanalysis.

As mentioned above, OBS-ONLY-CARRA is independent for CARRA-Land-Pv1, but not for CARRA, so it will not give a fair comparison. Indeed it serves to show the impact outside of observed locations seen by the assimilation. For that purpose we argue that one should compare with CTRL to evaluate the performance of the assimilation outside observation locations. In that case, scores are improved using the assimilation. Evaluation of the large scale improvement is also very difficult due to the representation error of the observations as discussed in the manuscript. Furthermore, the fact that CARRA-Land-Pv1 significantly improves in stations assimilated by both systems at least gives confidence in the method. We agree evaluation is a better suited word and change accordingly.

L307 It should be explained that CRPS is identical to MAE in the case of a deterministic forecast.

**We add an explanation for this in the revised manuscript**

L333-334 I would like to be optimistic but compared to independent observations (OBS-ONLY-CARRA) not assimilated in the offline reanalysis, the results do not exhibit improvements of snow depths (Fig 8c, Table 3) and exhibit significant a deterioration of density (Fig 7). I think a more qualified discussion of results is needed allowing a better questioning of the choices for data assimilation.

We understand the concern, however we do not fully agree that the results do not exhibit improvements of snow depths. First, we remind that the OBS-ONLY-CARRA set is assimilated in CARRA so we would expect CARRA to be significantly better in those locations, Figure 8 only shows small differences in MAE scores. While not independent, the OBS-BOTH set gives a fair comparison of where observations are available and the improvement in CARRA-Land-Pv1 vs CARRA is clear. That said, we agree this is not discussed sufficiently and needs to be strengthened in the revised manuscript.

L343-357 I think the discussion about the link between density and mass could be improved.

« We emphasize that these stations are not collocated with assimilated snow depth observations ». In a spatialized assimilation system, I think the goal is to improve snow cover at large scale, not only at the assimilated observations.

Yes, to some extent, but it is also a question about what observations actually represent and over how large scales they are useful. However, we revise the discussion about mass-density link.

L358 « By exploring the different observation sets used in CARRA and CARRA-Land-Pv1, we evaluate the analysis performance where observations are not available for assimilation ». Actually, only one dataset is independent and the results do not really focus on this one. I think the evaluation dataset used in this paper should be questioned and probably extended. A leave-one-out approach as in Cluzet et al., 2022 could be a valuable approach to consider all the snow depth datasets. Independent observations of satellite snow cover fractions could also be considered for a spatialized evaluation of the added value of data assimilation, especially in a context where such observations were assimilated by CARRA and are no longer assimilated in CARRA-Land-PV1.

We agree that none of the snow depth observations used are truly independent to document improvement relative to the CARRA reference. However, comparison against the CTRL using the OBS-ONLY-CARRA observations, which in this context is independent, does show consistent improvement.

L361 It is difficult to have a critical reading of this discussion due to the lacks in the methodology description. For instance, authors say that « the distance based localization used in CARRA-Land-Pv1 cause several stations in OBS-ONLY-CARRA to be unreachable in the analysis. C ». But the localization distance has not been defined neither provided, and the typical observation density is not known. I think a discussion should more rely on quantitative considerations provided to the reader.

We thank the reviewer for pointing this out, we add the missing description in the revised manuscript.

L384-391 Authors explain again the theoretical added value of their framework compared to the simple assimilation process of CARRA. However, my feeling is that they have not found the results illustrating how considering variable background uncertainties is able to improve assimilation compared to their reference benchmark. If authors have illustrations on specific cases, as suggested, I would encourage to present the material in the paper to better emphasize the potential added value, if it can not be demonstrated with finalized scores.

We agree that the method used cannot be concluded to always perform better than the CARRA based on the evaluation in this manuscript. However, we argue that the method is successful in updating the multivariate control vector which is not possible with the reference system. While it can not directly prove any benefit in unobserved areas, the method gives improved results where observations are available.

L393 implemented

**We correct this in the revised manuscript.**

L396-397 Again, this conclusion seems rather optimistic considering only independent evaluation datasets.

L399-401 Again, my feeling is that this theorical advantage is not completely supported by results.

We revise the conclusion considering these comments and with more criticism regarding the improvement of the proposed methods relative to CARRA.

---

## Author Comment (AC3)

**Response letter to CC1**

**Dear Nima Zafarmomen,**

We thank you for your comments on our manuscript, we believe that they help improve our manuscript. In the following, the comments are posted with our response in red text.

The paper addresses an important gap: bringing a flow-dependent ensemble Kalman framework to a multi-layer snow scheme for a high-latitude regional reanalysis. The topic is timely and the modelling chain (SURFEX–ISBA explicit snow + LETKF, driven by CARRA) is potentially valuable for cryospheric and hydrologic communities.

**Forcing and domain**

The manuscript states that CARRA forcing is "interpolated to the model grid" but omits grid spacing for both driver and land model. Domain limits are described in text but not in coordinates; include bounding box and spatial resolution.

We add the description of grid spacing in the text, and modify Figure 2 to cover the model area based on this comment.

**Ensemble generation**

Only perturbing atmospheric forcing inevitably under-represents uncertainty in snow compaction, albedo metamorphism, and interception. You should quantify how ensemble spread compares to innovation statistics (e.g., spread-skill ratio) to demonstrate sufficiency of the perturbation strategy. If spread is systematically low, adding multiplicative inflation alone is insufficient; process perturbations or parameter perturbations may be needed.

This is a valid concern and we agree that only perturbing forcing will not produce a large spread of the ensemble and possibly result in an over confident background state. In the manuscript we discuss how the perturbation strategy might suffer from this limitation, and that additional perturbations of the state or model parameters should be further developed. However, we argue that only perturbing the forcing data ensures that the relationship between model variables are consistent and governed by the model physics. Introducing perturbations to state variables and model parameters would require careful implementation, to avoid spurious correlations, and potentially a need for a significant increase in ensemble members to address this, which would also increase the computational demand. For a multi-year reanalysis system, we need to balance the computational cost vs the added benefit of a particular methodology. Regarding spread-skill ratio they can provide some insight in the ensemble quality. However, due to the large error of representation of in situ snow depth measurements, even a good ensemble could appear underspread. To provide the requested assessment, good estimates of observation representation errors also need to be quantified. We discuss this issue from L370 and suggest future studies on this particular topic.

The "remapping" approach for precipitation displacement is innovative, yet Appendix A lacks diagnostic evidence that the scheme produces realistic error structures. Provide at minimum a variogram or visual comparison between perturbed and reference precipitation fields.

In the revised manuscript we suggest updating Fig. 1R to show a field from the model derived precipitation and how this is remapped. The figure now gives a visual comparison between the perturbed and reference fields.

Figure 1R Precipitation field on model resolution with a small (3 member) ensemble. Upper left panel shows precipitation rate from the CARRA dataset, upper right and lower panels show ensemble realizations after remapping is applied. The star marks a reference point, for example an observation site.

**Ensemble size**

Ten members is very small for a 36-variable profile. You report that 20 members offered "no considerable degradation", but give no metrics. Include a sensitivity figure (e.g., CRPS vs. ensemble size) to justify the final choice.

We agree that this is a valid concern. In the literature, this has been a recurrent topic and "small" ensembles have shown to be sufficient for land surface data assimilation. As mentioned above, long reanalysis products need to prioritize computational cost when possible. We also want to argue that while the control vector is of size 36, many of the variables are strongly correlated, as shown in the profile figure (Fig. 5) in the manuscript. Furthermore, only five (forcing) variables are perturbed and the perturbations are cross-correlated.

Increment analysis (Sect. 3.1)

Figure 4 shows domain-mean increments of several millimetres water equivalent per day—this is large. Provide histograms or spatial standard deviations to make clear whether these increments are isolated to specific subregions or pervasive. Without that context, the reader cannot judge if the LETKF is "adding missing precipitation" or merely compensating biased forcing.

We appreciate this comment and agree that this is an important detail. Figure 4 in the manuscript shows that the increments, for all variables, are largest during the spring/melting phase. Whether this is because of precipitation (missed events or bias in amount) or due to the melting process, or due to other forcing variables (eg. too warm temperature) is not easy to determine. Below we include the requested increment standard deviation maps. The maps indicate that the larger increments are found over mountain areas, and that the areas of large mean increments (Fig. 3 in the manuscript) are less pronounced in terms of standard deviation.

Figure 2R: Standard deviation of increments for a) snow depth, b) snow water equivalent and c) snow density.

**Skill metrics**

- CRPS and MAE are reported, but no sampling uncertainty is provided. Bootstrap confidence intervals would show whether the apparent improvements are statistically robust.
- Station splits (OBS-ONLY-Pv1, etc.) prove useful, yet the sample sizes differ dramatically.
  Present RMSE normalised by climatological variance to avoid overweighting dense station clusters.

The metrics presented in Table 3 were recomputed using bootstrap resampling. The resulting 95% confidence intervals (2.5th–97.5th percentiles) indicated statistically significant differences between the compared values, as the intervals did not overlap.

**SWE validation**

Only six pillow sites are available, but you can still compute Kling-Gupta or Nash–Sutcliffe across time to give hydrologists a sense of hydro-logical skill. Also, the negative bias at one degraded site coincides with orographic precipitation maxima; examine whether forcing under-catch is the root cause.

The figure below presents Kling-Gupta Efficiency scores for each snow pillow station. These results reflect the same relative performance patterns already illustrated in Figure 10 of the manuscript. While the figure may be of interest to some readers, we have chosen not to include the Kling-Gupta values in the revised paper, as they only provide limited additional insight beyond the metrics already discussed.

Figure 3R: Kling-Gupta scores for CARRA-Land-Pv1 (ANA) vs the reference datasets (CARRA and CTRL)

I strongly recommend that the authors expand their discussion, as data assimilation is not only applicable to snow schemes but is also widely used in other areas such as streamflow prediction. I also recommend citing the following papers: Optimising ensemble streamflow predictions with bias correction and data assimilation techniques; Assimilation of Sentinel-Based Leaf Area Index for Modeling Surface-Ground Water Interactions in Irrigation Districts